# Hydrogel-in-hydrogel live bioprinting for guidance and control of organoids and organotypic cultures

Anna Urciuolo [1,2,11] ✉, Giovanni Giuseppe Giobbe [3,11], Yixiao Dong[4,11], Federica Michielin[3], Luca Brandolino[5,6], Michael Magnussen [3], Onelia Gagliano [5,6], Giulia Selmin[3], Valentina Scattolini [2], Paolo Raffa[2], Paola Caccin[7], Soichi Shibuya [3], Dominic Scaglioni [3], Xuechun Wang[4], Ju Qu[4], Marko Nikolic [3], Marco Montagner[1], Gabriel L. Galea [3], Hans Clevers [8,10], Monica Giomo [5], Paolo De Coppi [3,9] & Nicola Elvassore [3,5,6] ✉

Three-dimensional hydrogel-based organ-like cultures can be applied to study development, regeneration, and disease in vitro. However, the control of engineered hydrogel composition, mechanical properties and geometrical constraints tends to be restricted to the initial time of fabrication. Modulation of hydrogel characteristics over time and according to culture evolution is often not possible. Here, we overcome these limitations by developing a hydrogel-in-hydrogel live bioprinting approach that enables the dynamic fabrication of instructive hydrogel elements within pre-existing hydrogel-based organ-like cultures. This can be achieved by crosslinking photosensitive hydrogels via two-photon absorption at any time during culture. We show that instructive hydrogels guide neural axon directionality in growing organotypic spinal cords, and that hydrogel geometry and mechanical properties control differential cell migration in developing cancer organoids. Finally, we show that hydrogel constraints promote cell polarity in liver organoids, guide small intestinal organoid morphogenesis and control lung tip bifurcation according to the hydrogel composition and shape.

Three-dimensional (3D) organ-like cultures, such as organoids and ex-vivo organotypic cultures, are able to self-organize and recapitulate structures and some functionalities of specific tissues, thus providing physiologically relevant in vitro models for the study of organ development, regeneration and diseases[1,2]. The biological self-organization of organ-like cultures arises from progressive local interactions between cells in an initial environment. This is amplified by positive and negative feedback from biomechanical and biochemical factors, which steer the cellular self-organization in a highly context-dependent manner. Indeed, molecular and environmental fluctuations within specific time windows profoundly influence the final multicellular self-organization process[3]. Shaping such complex and

[1]Dept. of Molecular Medicine, University of Padova, Padova, Italy. [2]Istituto di Ricerca Pediatrica, Città della Speranza, Padova, Italy. [3]GOSICH Zayed Centre for Research into Rare Disease in Children, University College London, London, UK. [4]Shanghai Institute for Advanced Immunochemical Studies (SIAIS), ShanghaiTech University, Shanghai, China. [5]Dept. of Industrial Engineering, University of Padova, Padova, Italy. [6]Veneto Institute of Molecular Medicine, Padova, Italy. [7]Dept. of Biomedical Science, University of Padova, Padova, Italy. [8]Hubrecht Institute, KNAW and University Medical Center, Utrecht, The Netherlands. [9]Dept. of Specialist Neonatal and Paediatric Surgery, Great Ormond Street Hospital, London, UK. [10]Present address: Pharma Research and Early Development (pRED) of Roche, Basel, Switzerland. [11]These authors contributed equally: Anna Urciuolo, Giovanni Giuseppe Giobbe, Yixiao Dong. ✉e-mail: anna.urciuolo@unipd.it; nicola.elvassore@unipd.it

chaotic systems in space and time has proven extremely challenging so far. Recent efforts to overcome this limitation have combined organogenesis knowledge and bioengineering-inspired culture systems[4], with a particular focus on hydrogel-based approaches[5]. As an example, organoid experimental traceability, reproducibility and physiological relevance have recently been improved by controlling the initial environmental culture conditions via hydrogel-based technology[6–8]. However, the composition, mechanical properties, and geometrical constraints of the engineered 3D scaffolds are pre-designed and set at the initial state of cell culture. Thus, existing approaches do not allow precise modulation of hydrogel shape and physical properties over time, corresponding to the morphological modifications that take place during cell culture.

We have previously shown that intravital 3D bioprinting can be applied in vitro to impose initial 3D constraints on intestinal organoid growth[9]. We demonstrated that hydrogel fabrication can be achieved within Matrigel droplets (a gold standard hydrogel system for organoid culture) concomitantly with cell seeding. To do so, a photo-sensitive polymer was pre-mixed with liquid Matrigel and the mixed solution was gelated before cross-linking the polymer through two-photon (2P) mediated bioprinting[9]. We showed that 7-hydroxycoumarin-3-carboxylic acid (HCCA) and 7-carboxymethoxy-4-methylcoumarin (CMMC) conjugated with polymers, such as branched PEG and gelatin, respectively 7-Hydroxycoumarin-3-carboxylate-PEG (HCC-PEG) and HCC-Gelatin or CMMC-PEG and CMMC-gelatin, can be efficiently crosslinked by 2P microscopy irradiation at the moment of cell seeding for organoid culture generation[9]. However, a limitation of our previous work was the impossibility of dynamically controlling the imposed architecture during the evolution of organ-like culture size, shape, and cell identity, including emerging specialized cell types.

Here, we developed a method to dynamically fabricate 3D hydrogel structures within hydrogel-based organ-like cultures. Newly fabricated hydrogels can now be tailored to the specific culture system requirements and at the desired culture time, matching the conditions necessary to dynamically control 3D organ-like cultures. This technique allows the fabrication of natural or synthetic hydrogels that scale from sub- (10 micrometer) to supra-organoid (millimeter) sizes with desired 3D shapes and tunable mechanical properties able to guide and control cell behavior of 3D organ-like cultures. To assess the performance of this methodology on different culture conditions, we used organoids, including human liver, mouse intestinal and cancer organoids, as well as organotypic cultures of spinal cord and lung in Matrigel and collagen hydrogels. This provides an extensive validation of the proposed technology and addresses specific features of the experimental demands. Since this technology combines live imaging with real time bioprinting, from now on we refer to it as hydrogel-in-hydrogel live bioprinting.

## Results

### Photosensitive hydrogels can be crosslinked via 2P mediated printing within pre-existing hydrogels

In order to dynamically control the organ-like cultures, we hypothesized that a photo-sensitive liquid polymer could be loaded within the pre-existing gelated Matrigel at the desired cell culture time point and subsequently crosslinked into 3D hydrogel structures. This is based on the idea that pre-existing solid hydrogel might allow free diffusion of soluble photo-sensitive polymers within their volume. Subsequently, photo-sensitive polymer crosslinking via femtosecond near-infrared tightly focused pulsed laser irradiation would enable the fabrication of 3D hydrogel-in-hydrogel structures (Fig. 1a). In principle, this strategy should allow the crosslinking of natural or synthetic photosensitive polymers loaded into the 3D organ-like culture with complex geometry, additive manufacturing, and tunable mechanical properties via 2P-microscope irradiation. Importantly, the temporal control of

hydrogel-in-hydrogel fabrication can be simultaneously combined with live imaging. This allows for precise positioning and orientation of the fabricating 3D hydrogel relative to a specific organoid, to a defined portion of the organoid, or to the ex-vivo organotypic culture.

To test this hypothesis, we first investigated the self-diffusion coefficient of polymers in solid Matrigel droplets, which is the most commonly used natural hydrogel for 3D organoid culture. Fluorescence Recovery After Photobleaching (FRAP) analyses showed that fluorescein Isothiocyanate-Dextran (FITC-dextrans) with different molecular weights freely diffused within solid Matrigel according to Fick's law, with a diffusion coefficient equal to $2.5 \pm 0.4 \, \mu m/s^2$ or $1.4 \pm 0.1 \, \mu m/s^2$ for 40 or 500 kDa molecular weight FITC-dextrans respectively (Fig. 1b). These diffusivities are of the same order of magnitude of polymer in water and enabled FITC-dextrans to reach saturation in <300 s (5 min) within Matrigel droplets of 100 μL.

We next tested whether diffused soluble photo-sensitive polymers could subsequently be crosslinked into solid structures. We added liquid HCC-Gelatin on top of pre-existing hydrogel droplets (Matrigel) and incubated for 15 min before performing 3D printing. After incubation, HCC-Gelatin structures were successfully fabricated within the Matrigel according to software design as shown in Fig. 1c. Importantly, the hydrogels showed structural integrity over the 2 days of incubation of the printed Matrigel droplet in DMEM as confirmed by imaging analyses. We probed the efficiency and the accuracy of 3D hydrogel-in-hydrogel photo-crosslinking by fabricating hydrogel parallelepipeds of different heights and at multiple Z positions (Fig. 1c). Moreover, we also showed that hydrogels with minimal linewidth ($1.5 \pm 0.8 \, \mu m$) can be fabricated as linear objects with single line scan and freeline scan program (Fig. 1d). A resolution of 3 μm between two scan lines was achieved (Fig. 1e). Accordingly, a 3D single line spiral-shaped hydrogel of HCC-Gelatin within Matrigel was fabricated using a computer drawing pad combined to a free line scan program of the multi-photon microscope (Fig. 1f).

The reproducibility of hydrogel-in-hydrogel photo-crosslinking and stability of the fabricated structures was confirmed by the absence of swelling (Fig. 1g) and the preservation of the Young's modulus of HCC-Gelatin hydrogels printed within Matrigel just after (D0) or 2 days after Matrigel gelation (D2) (Fig. 1h). Importantly, we showed that hydrogel-in-hydrogel 3D bioprinting does not damage living cells and can be performed in close proximity of human hepatic organoids[9–11] (Supplementary Fig. 1a). To do so, HCC-Gelatin loaded within the 3D organoid culture was crosslinked into the desired 3D shape and tailored on individual selected organoids, preserving the overall organoid morphology (Supplementary Fig. 1b,c).

Overall, these data demonstrate that hydrogel-in-hydrogel 3D bioprinting of photosensitive polymers can be efficiently achieved.

### Hydrogel-in-hydrogel live bioprinting of organotypic spinal cord (oSpC) 3D cultures allows the control of neural axon sprouting

We challenged the hydrogel-in-hydrogel live bioprinting versatility in a complex 3D cell culture system, ie: ex-vivo 3D organotypic spinal cord cultures. Ex-vivo 3D organotypic spinal cord cultures are a suitable model to investigate the potential of hydrogel-in-hydrogel fabrication in a dynamic event such as 3D neural axon sprouting[12]. Such organotypic cultures retain the multicellular and structural complexity of the native tissue[13,14], and can be maintained in 3D culture for up to 14 days to study neural axon sprouting[12]. Spinal cord sections were derived from fetal rats, embedded in Matrigel droplets and cultured for 2 days from when neural axons started sprouting from the central body of the organotypic section (Supplementary Fig. 2a). At this stage −i.e. 7 days of culture, live bioprinting was performed to fabricate HCC-Gelatin hydrogels (Fig. 2 and Supplementary Fig. 2). We first investigated whether the presence of neural projections and/or central body of the oSpC could interfere with hydrogel bioprinting. To do so, parallelepiped-shaped structures were designed to be located above

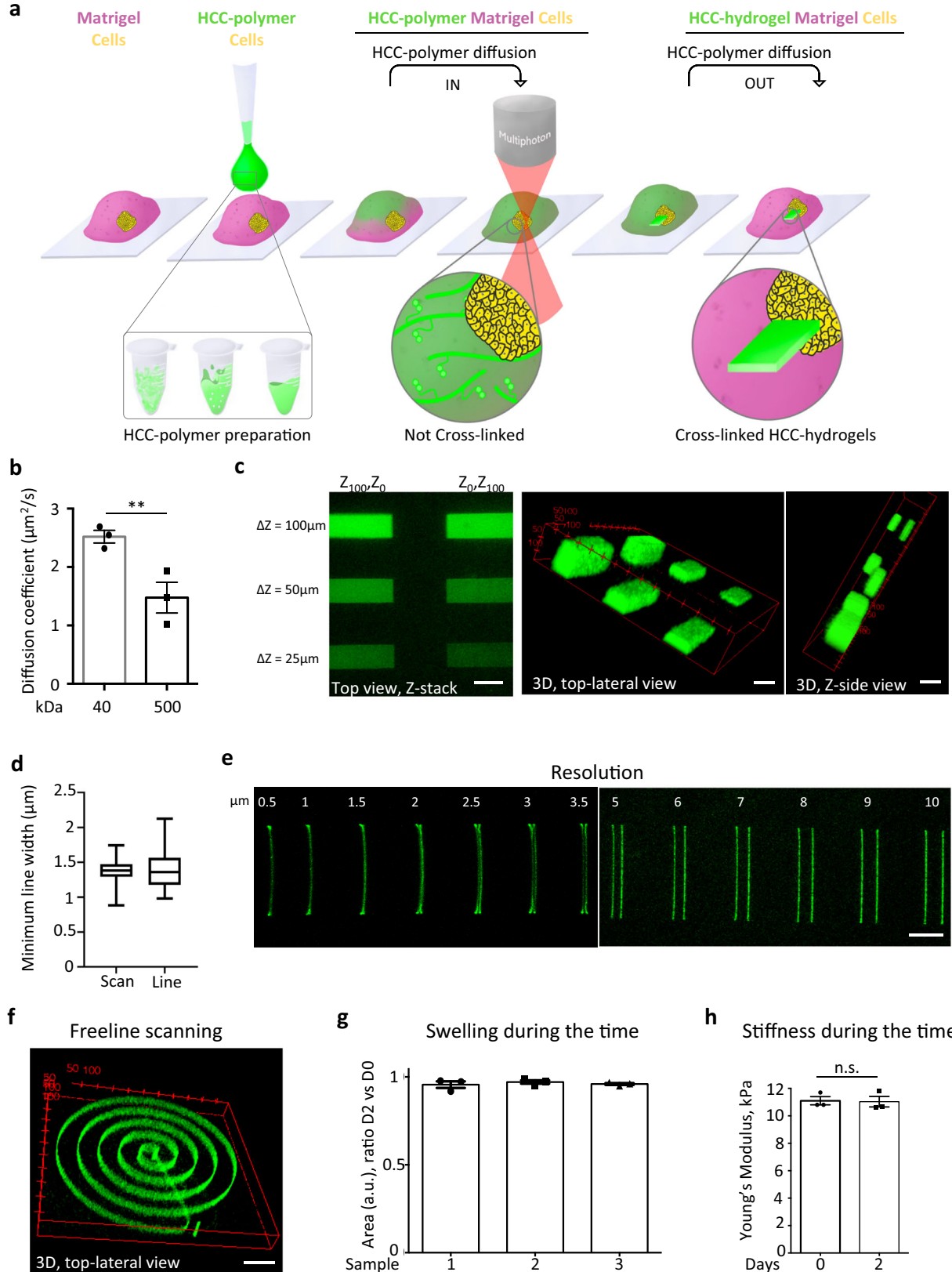

the central body of the oSpC and in proximity of extruded neural projections. We observed that hydrogels were correctly fabricated above the oSpCs, without interference of the laser path across the central body of the oSpC, nor the neural projection (Fig. 2a). Moreover, as shown in Fig. 2b, c, multiple structures can be printed at different Z planes to embed cellular projections within the hydrogel without affecting cell vitality and integrity (Fig. 2b, c). Parallel results were achieved when linear objects were generated with freeline scan mode irradiation across neural projections (Fig. 2d). Finally, to test whether the hydrogel could control oSpC cellular projection outgrowth, HCC-Gelatin parallelepiped-shaped structures were perpendicularly oriented in respect to the central body of the organotypic section pre-

**Fig. 1 | Performance of hydrogel-in-hydrogel printing. a** Strategy and set-up for hydrogel-in-hydrogel live bioprinting. HCC-hydrogel 2P-printing can be performed within solid gel of a 3D organ-like culture at any experimentally required time point of cell growth by (i) allowing liquid HCC-polymers to diffuse within the pre-existing solid gel, (ii) fabricating 3D hydrogel objects by using a multiphoton microscope equipped with a motorized xyz stage and a femtosecond near-infrared tightly-focused pulsed laser emission, (iii) removing the un-crosslinked HCC-polymers from the 3D organ-like culture via diffusion. **b** Quantification of the diffusion coefficient of 40 (grey) or 500 (black) kDa FITC-dextrans within Matrigel. Data are shown as mean ± s.d. of three independent replicates; unequal variance Student's *t*-test; *$P < 0.0213$. **c** Left, representative confocal z-stack images of sequentially fabricated HCC–gel hydrogels (green) within the same Matrigel drop and printed at different xyz positions by using near-infrared laser pulses through a multiphoton microscope; total $\Delta z = 100\,\mu m$ or $50\,\mu m$ or $25\,\mu m$. Scale bars, $100\,\mu m$. Middle and right, 3D-volume reconstruction reveals the volumetric position of the various objects; coordinates are shown in red. **d** Quantification of the minimum line width

obtained using scan or freeline scanning mode, respectively scan and line scan. Data are shown as mean ± s.d. of three independent replicates. **e** Multiple HCC–Gel structures of three independent replicates fabricated by near-infrared multiphoton laser pulses. $\Delta z = 20\,\mu m$. Scale bar, $20\,\mu m$. **f** Representative 3D reconstruction of three independent replicates of a HCC–Gel spiral-shaped hydrogel fabricated using the free line-scan mode. $\Delta z = 30\,\mu m$. Coordinates are shown in red; scale bar, $100\,\mu m$. **g** Quantification of the area of hydrogels sequentially fabricated at fixed laser power (1 mW) and wavelength (800 nm) within the same Matrigel drop; each hydrogel series was analyzed just after photo-crosslinking (day 0) or at 2 days after the last 3D bioprinting. Data are shown as mean ± s.d. of three independent replicates. **h** Young's modulus measured by atomic force microscopy of hydrogels photo-crosslinked at fixed laser power (1 mW) and wavelength (800 nm). Each hydrogel series was analyzed just after photo-crosslinking (day 0) or at 2 days after the last 3D bioprinting. Data are shown as mean ± s.d. of three independent replicates; multiple comparison one-way ANOVA was used; n.s., not statistically significant.

cultured for 2 days (Supplementary Fig. 2a). Samples were cultured for a further 2 days–i.e. 4 days of culture, and neural axon sprouting was evaluated by imaging analysis (Fig. 2e, f and Supplementary Fig. 2b). Interestingly, neural axons showed a patterned organization only where the HCC-Gelatin hydrogels were present and according to the hydrogel shape (Fig. 2e). Quantification of neural axon directionality[12] shows that hydrogel-in-hydrogel live bioprinted structures guided neural axons alignment, in comparison to the randomly organized neural projections located within the 3D space of the untreated volume of the Matrigel droplet (Fig. 2f). Result is parallel arrays of predictably directional axons amenable to imaging-based analyses.

Overall, we showed that hydrogel-in-hydrogel bioprinting can be performed in growing oSpC without damaging neural projections, and be used to control the directionality of neural axon spouting.

### Dynamic fabrication of hydrogels reveals differential ability of cancer cells to migrate in 3D environment

Subsequently, we challenged our approach to serve as an innovative method to study how dynamic changes in tissue architecture shape the behavior of cancer cells in 3D in vitro systems, such as tumor spheroids. Current engineered approaches cannot be adapted to varying topology and space surrounding cancer cells over time. Consequently, paths and architecture are often established at the beginning of the experiments. Here, we cultured A549 lung adenocarcinoma-derived organoids to evaluate the effect of dynamically imposed geometrical constraints on organoid growth and cancer cell migration. Moreover, we tested whether the hydrogel-in-hydrogel live bioprinting platform can be expanded to other biomaterials by using non-cell adhesive CMMC-PEG hydrogels in combination with collagen-based 3D cultures. Initially, 1 day after organoid culture, a series of 3D CMMC-PEG pillars distanced $10\,\mu m$ from each other where fabricated within the pre-existing hydrogel to confine the selected organoid (Supplementary Fig. 3a). Cell viability was confirmed after the bioprinting of pillars around the organoids (Supplementary Fig. 3b, c). Live imaging analysis showed that during growth, cancer organoids first contacted the fabricated hydrogels and then subsequently started deforming the HCC-PEG pillars until the cancer cells could overcome the geometrical constraint given by the pillars (Fig. 3a and Supplementary Video. 1). Actin staining analysis showed that first cell projections and, after that, the entire cellular body can migrate from the organoid between the pillars (Fig. 3b).

To assess whether the hydrogel deformability could influence cancer cell migration, we used CMM-4arm or CMM-8arm PEG polymers to bioprint pillars with different mechanical properties around organoids. CMM-4arm PEG hydrogels showed statistically significant lower stiffness when compared to CMM-8arm PEG hydrogels (Fig. 3c). Young's modulus differences between the two PEG-based hydrogels were preserved also by varying the laser power used for cross-linking (Fig. 3c). Interestingly, we observed different migration of cells from

organoids confined by the 4- as opposed to 8-arm PEG pillars. In particular, a significantly higher number of nuclei were identified outside the more deformable pillar barrier (4-arm PEG) 7 days after printing (8 days from cell seeding) (Fig. 3d,e). We then fabricated a first array of pillars distanced $40\,\mu m$ from each other to confine a specific cancer organoid (Fig. 3f and Supplementary Fig. 3d). As soon as the growing organoid contacted the pillars (day 6 after first bioprinting), an additional hydrogel-in-hydrogel bioprinting was performed to generate a cage of CMMC-PEG pillars distanced $15\,\mu m$ from each other surrounding the organoid together with the previously fabricated CMMC-hydrogels (Fig. 3f and Supplementary Fig. 3d). In agreement with the previous results, this second step of bioprinting did not impinge on cell viability (Supplementary Fig. 3e). However, in contrast to the previous results, live imaging analysis showed that after contacting the first array of distanced bioprinted pillars, growing organoids modified their shape, embracing the fabricated pillars (Fig. 3f, g). As soon as the cancer organoid reached the second array of closer pillars, migratory events could be observed, as shown by actin imaging analysis of cell protrusion across the imposed geometrical constraints (Fig. 3h).

Altogether this data demonstrates that 3D structures with different size and geometry can easily be generated in the proximity of tumor spheroids to investigate how evolving confinements and tracks in 3D culture can affect cancer cell migration dynamic.

### Hydrogel-in-hydrogel live bioprinting can be used to shape small intestinal organoid morphogenesis

We then challenged the possibility to use our hydrogel-in-hydrogel bioprinting technology to generate structures with supra-organoid size capable of guiding organoid morphogenesis. For this aim, we took advantage of the LGR5-EGFP-DTR mouse small intestinal organoids (mSIOs)[15] where Lgr5+-stem cells are fluorescently labeled within the crypt compartment separated from the differentiated villus portion[16,17]. This reporter cell line allows direct evaluation of how geometric constraint alters organoid morphogenesis and mSIOs budding. First, experimental optimization showed that HCC-Gelatin and non-adhesive HCC-hydrogel (PEG based)[9] structures impose a constraint on growing cystic mSIOs inducing budding and, from day 2 to day 7 of culture, mSIOs LGR5 cell segregation (Supplementary Fig. 4a-c).

Then, we designed a complex supra-organoid 3D structure to mimic primordial small intestine development[18]. This structure was characterized by 3 crypt-shaped regions per single $50\,\mu m$ z-plane (height), with an anticlockwise rotation of 60° and 30° every subsequent z-plane (Fig. 4a and Supplementary Fig. 5a). The three-dimensional design was faithfully reproduced with the printed gelatin structures, as shown by autofluorescence and bright field images (Fig. 4a, Supplementary Fig. 5a and Supplementary Video. 2). We then transferred such design to 3D mSIO culture, enclosing a single mSIO within the internal lumen of the bioprinted hydrogel 1 day after cell

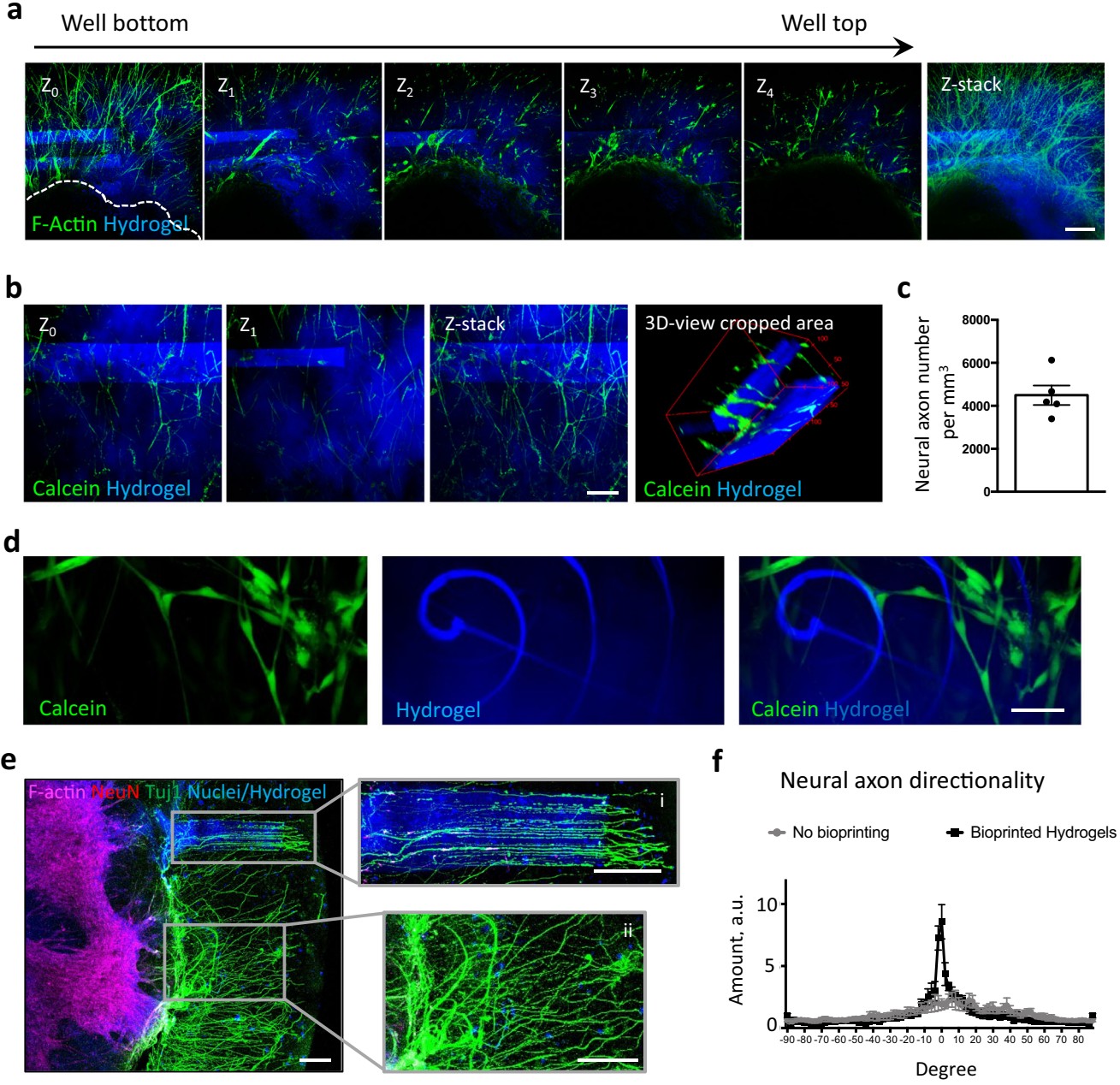

**Fig. 2 | Temporal control of hydrogel-in-hydrogel live bioprinting and controlled axon guidance of oSpCs. a** Confocal fluorescence images acquired at different ΔZ ($Z_0$, $Z_1$, $Z_3$, $Z_4$) and relative Z-stack imaging analysis showing the presence of hydrogels (blue) fabricated above the central body of the oSpC cultured in 3D matrigel droplet for 7 days before bioprinting. Phalloidin (green) was used to detect cellular projections. Scale bar, 100 μm. **b** Confocal fluorescence images acquired at different ΔZ ($Z_0$, $Z_1$) and relative Z-stack imaging analysis showing the presence of multiple hydrogels (blue) fabricated at different Z planes that embed alive (calcein-positive, green) cellular projections of the oSpC cultured in 3D matrigel droplet for 7 days before bioprinting. Scale bars, 100 μm. **c** Quantification of vital (calceine-positive) neuronal projections embedded within bioprinted hydrogel volume. Data are shown as mean ± s.d. of five independent replicates; unequal variance Student's *t*-test was used; *P* < 0.05 was considered statistically significant. **d** Representative images showing integrity of single-line bioprinted hydrogel (blue) and vital (calcein-positive, green) cellular projection of the oSpC cultured in 3D matrigel droplet for 7 days before bioprinting. Scale bar, 10 μm. **e** Representative fluorescence images of a spinal cord culture showing alignment of axons protruding within fabricated hydrogel (i) as opposed to randomly oriented axon organization in absence of the hydrogel (ii). Scale bar, 200 μm. **f** Quantification of neural projection directionality performed in area where the neural projections were far (no bioprinting) or in proximity (bioprinted hydrogels) of the fabricated hydrogel-in-hydrogel structures.

seeding. We assessed cell viability 3 or 7 days after printing, with comparable results to the unprinted control mSIOs (Supplementary Fig. 5b). We followed the evolution of mSIO morphogenesis inside the HCC structure over days (Fig. 4b and Supplementary Fig. 5c). Induced mSIO budding took place in correspondence with the multiple 3D printed crypt shapes, with protrusion and invasion of LGR5 cells into the imposed HCC-Gelatin apertures (Fig. 4c and Supplementary

Fig. 5d). mSIOs adapted their shape according to the available space of the primordial intestine-shaped hydrogels during the days of culture

Structural stability of the primordial intestine-shaped hydrogel was confirmed by the absence of statistically significant difference in swelling during the days of culture (Supplementary Fig. 5e). To test whether we could control the ability of mSIO budding in such structures, we quantified incremental size changes of the mSIOs within the

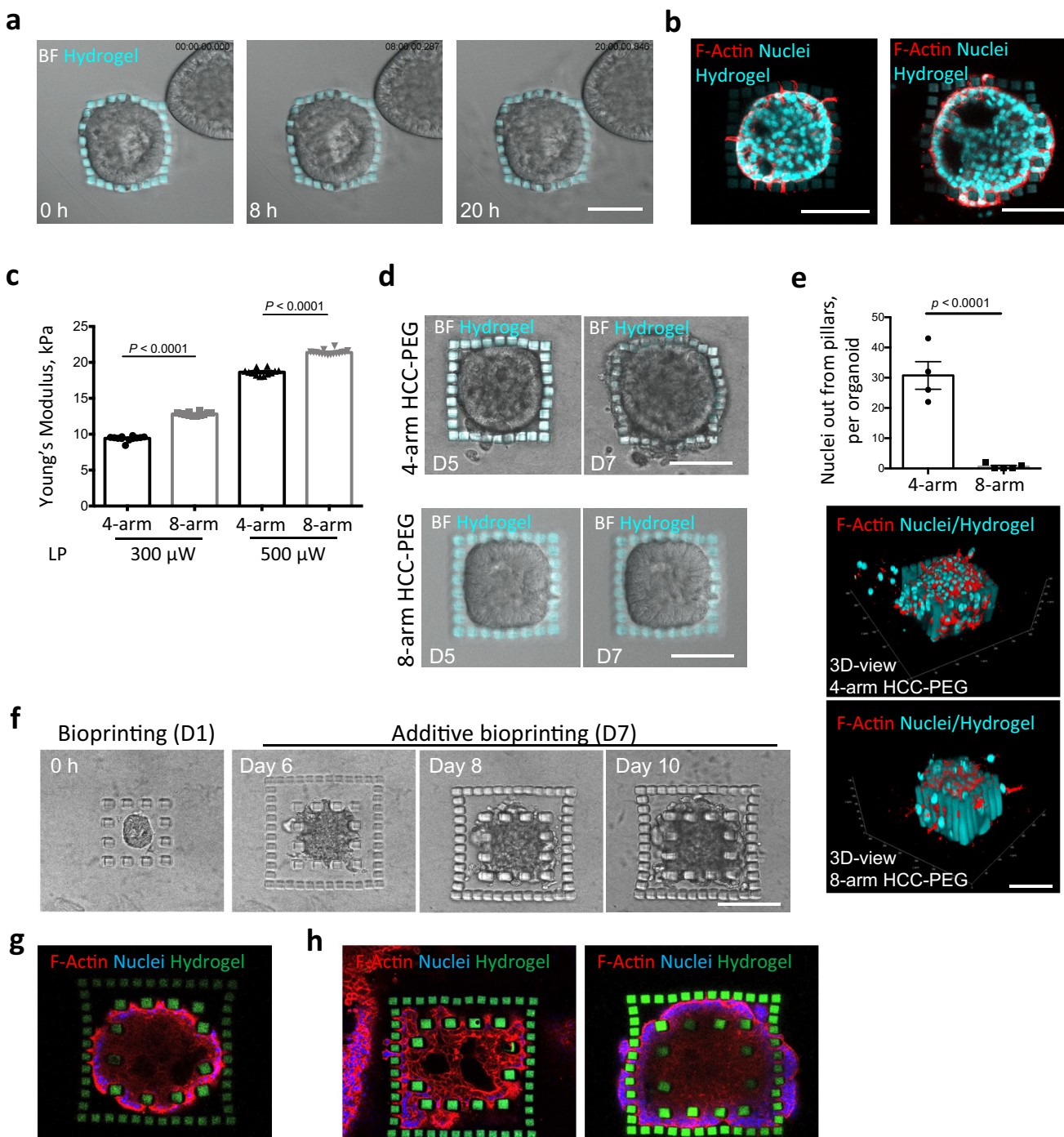

**Fig. 3 | Hydrogel-in-hydrogel live bioprinting for studying cancer cell migration in organoid 3D cultures. a** Brightfield time lap images (0, 8, 20 h) of a hydrogel-embedded tumor spheroid (day 6 post printing) growing within a cage of HCC-PEG pillars fabricated 1 day post organoid culture. Scale bar, 100 μm**. b** Fluorescent images of the tumor spheroid in **c**, showing different stages of cellular migration through the pillars day 7 post printing. Scale bars, 100 μm**. c** Young's modulus measured by atomic force microscopy of 4-arm (black) or 8-arm (gray) HCC-PEG hydrogels photo-crosslinked at increasing laser power (300 or 500 μW) and wavelength (800 nm). Hydrogels were sequentially fabricated within the same Matrigel drop. Data are shown as mean ± s.d. of three independent replicates; all measurements performed are reported; unequal variance Student's *t*-test was used; $P < 0.05$ was considered statistically significant. **d** Brightfield time lapse images of hydrogel-embedded tumor spheroid 5 or 7 days after bioprinting of 4-arm (upper panels) or 8-arm (lower panels) HCC-PEG pillars fabricated 1 day of organoid

culture. Scale bars, 100 μm. **e** Quantification of nuclei detected out of the pillars of the hydrogel-embedded tumor spheroid 7 days after bioprinting of 4-arm (black) or 8-arm (gray) HCC-PEG pillars fabricated 1 day of organoid culture. Representative images of 3D reconstructions with *xyz* coordinates and 50 μm scale bar are shown. **f** Four-arm HCC-PEG hydrogel pillars were fabricated 1 day or 7 days post organoid culture (days 0 and 6 post printing, respectively) around a hydrogel-embedded growing tumor organoid. The brightfield images show the growth of the caged tumor spheroid at different time points. Scale bar, 100 μm. **g** Representative fluorescent image of tumor spheroid as in **e** at 14 days from first bioprinting step (day 15 of organoid culture). Scale bar, 100 μm. **h** Representative fluorescent images of cancer cells protruding through the bars of the first bioprinted hydrogel cage, invading the surrounding space and migrating through the pillars of the second fabrication step. Scale bars, 100 μm.

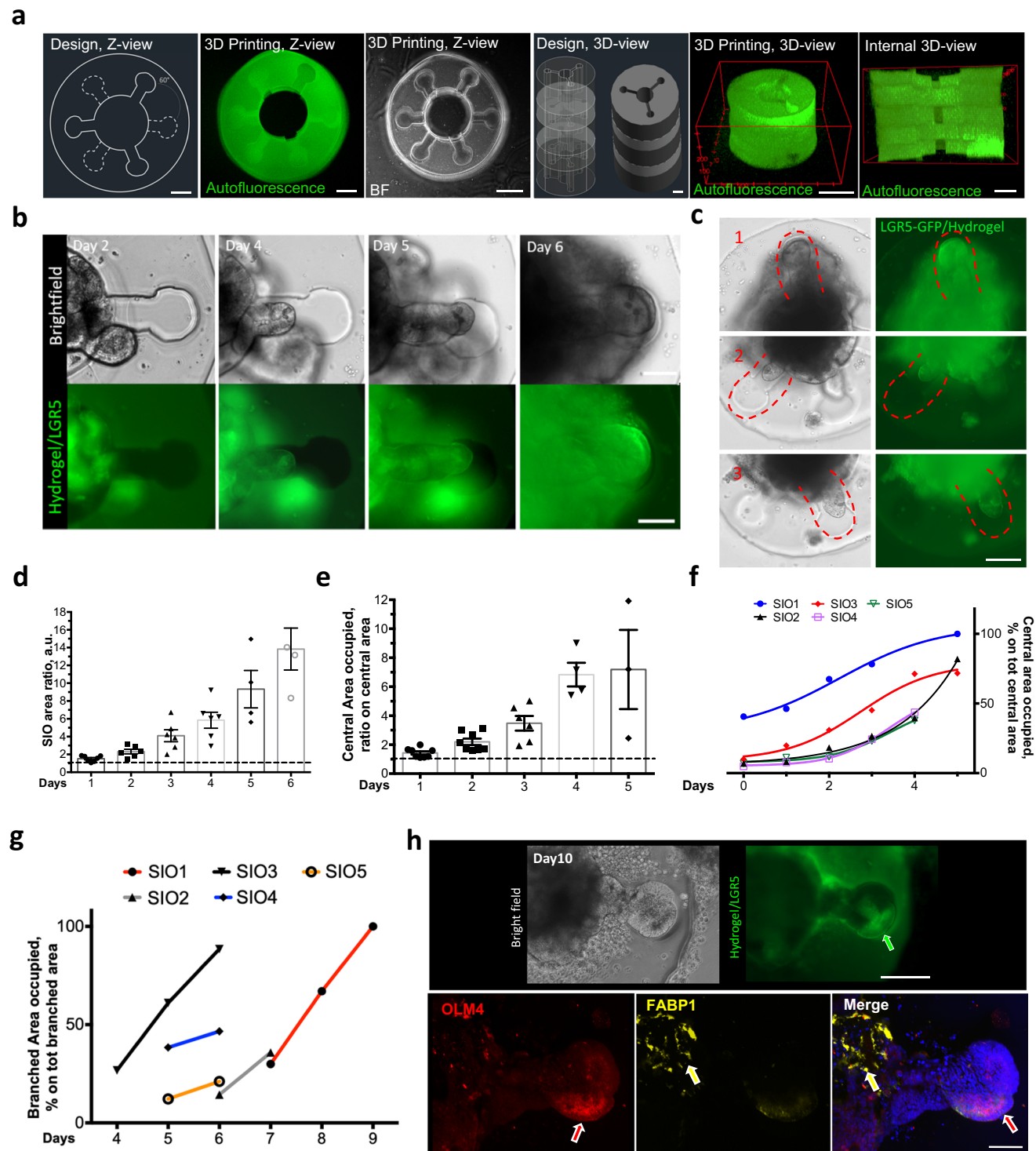

available space of the primordial intestine-shaped hydrogels (Fig. 4d). From the quantification of the central area occupied during the culture, we could assess that mSIOs were growing and adapting their shape to the central part of the hydrogels (Fig. 4e,f). We quantified in the same way the capacity of the mSIOs to invade the crypt-shaped portion of the primordial intestine-shaped hydrogel during culture (Fig. 4g). We then investigated the cell differentiation of the mSIOs after 10 days of culture in primordial intestine-shaped hydrogels. Staining for chromogranin A and lysozyme, markers of enteroendocrine and Paneth cells, respectively, confirmed that cellular identity was maintained (Supplementary Fig. 5f). Importantly, we observed specific expression of crypt stem cell markers olfactomedin 4 and

LGR5-GFP only in the crypt-shaped compartment of the primordial intestine hydrogel. Conversely, villus differentiation was restricted to the luminal region of the structure, as confirmed by the fatty acid-binding protein 1 staining (Fig. 4h). These data show that hydrogel-in-hydrogel live bioprinting can be tailored at supra-organoid size level to guide small intestinal organoid morphogenesis and differentiation with high temporal and spatial resolution.

## Hydrogel-in-hydrogel live bioprinting induces cell polarization of liver organoid 3D cultures

Based on the ability of geometrical constraints to control organ and organoid morphogenesis and biological cell responses[19], we

**Fig. 4 | Supra-organoid driven intestinal organoid morphogenesis via hydrogel-in-hydrogel live bioprinting. a** Representative bright field and fluorescence images showing 60°primordial small intestine design and HCC-gel hydrogels (left panel, top view; right panel, 3D reconstruction view). Scale bars 200 μm. **b** Representative bright field (upper) and fluorescence (lower) images of mSIOs just after primordial small intestine HCC-gel hydrogel printing or 2, 4, 5 or 6 days of culture and bioprinting. Budding was observed according to the defied shape of the hydrogel. Scale bars 100 μm. **c** Representative bright field and fluorescence images showing mSIO buds invading multiple crypts at different Z-levels of the primordial small intestine design after 6 days of culture post-printing. Scale bar 200 μm. **d** Quantification of the ratio between the area of the organoid at day 0 of culture (dashed line) and the area of the organoid during the following culture days (1–6 days). Statistical analysis is shown in Supplementary Table 1. **e** Ratio between the central area of the primordial intestine-shaped hydrogels with the organoid at seeding time (dashed line) and the area occupied by the organoid from day 1–6 of culture. Statistical analysis is shown in Supplementary Table 2. **f** Quantification of the percentage of central area occupied by the mSIOs during the culture (0–6 days). Calculation of the percentage was shown for 5 independent mSIO cultures. **g** Quantification of the percentage of branched areas occupied by 5 independent mSIO during the cultures (4–9 days). **h** Upper panels, representative bright field (upper) and fluorescence (lower) images showing mSIO budding after 10 days of culture within the primordial small intestine-shaped HCC-gel hydrogel. The arrow points at the LGR5 (green) cells. Lower panels, representative images showing immunofluorescence analysis for OLM4 (red) (corresponding to the LGR5-GFP in (**i**) and FABP1 (yellow) of mSIO cultured for 10 days within the primordial small intestine-shaped HCC-gel hydrogel. Nuclei are stained with Hoechst (blue). The arrows point at the branched (red) or central (yellow) portion of the mSIO in respect to the hydrogel. Scale bars 100 μm (upper panels), 50 μm (lowe panels).

investigated if hydrogel-in-hydrogel live bioprinting could be used to control hepatocyte polarization[20] in human fetal hepatocyte organoid 3D cultures. Mechanically disaggregated hepatocyte organoids were seeded in Matrigel, and live bioprinting was performed to generate HCC-Gelatin hydrogels as distant walls or adjacent pillar structures around the forming organoids (Fig. 5a,b). The samples were kept in expansion medium for 6 days and differentiated for a further 24 h. Imaging analysis showed preservation of hydrogel integrity and human liver organoid morphology after bioprinting (Fig. 5a,b). Organoids grew and remodeled their shape according to the constraints imposed by the bioprinted structures (Fig. 5a). Interestingly, in the presence of HCC-Gelatin hydrogels, but in absence of physical constraints, i.e. distant walls, no apical-basal polarization of the cells was observed, as shown by staining for multidrug resistance-associated protein 2 (MRP2) and zonula occludens-1 (ZO-1) (Fig. 5b). Strikingly, we observed that cells interacting with HCC-gelatin pillars polarized with a marked expression of MRP2 and ZO-1 towards the internal portion of the organoid, while β4-integrin (INTβ4) remained confined on the basal layer (Fig. 5b). Some polarized behavior was observed when the walls were printed adjacent to the growing organoid (Supplementary Fig. 6).

This data further confirmed that hydrogel-in-hydrogel bioprinting can be applied to target sub-organoid structures to induce organoid morphogenesis, as apical-basal polarization of the cells in liver organoids.

### Lung tip bifurcation can be controlled in 3D cultures of lung rudiments by imposing geometrical constrains

Next, we investigated the effect of live bioprinting on ex vivo organotypic cultures of mouse fetal mesenchyme-free lung epithelium rudiments (Fig. 5c). We coupled the 3D organotypic culture of lung epithelium rudiments in Matrigel drops with the photo-printing of 8-arm HCC-PEG structures to control the direction of spontaneous branching, under optimized culture conditions[21]. Specifically, circular pillars were printed after 24 h of 3D lung rudiment culture in close proximity to bud tips and branching was monitored during the first cycle of epithelial budding (Fig. 5d). Time-lapse imaging over 3 days with 5-h intervals shows that physical interaction of the fetal lung tips with the pillar drove bud tip bifurcation (Fig. 5d and Supplementary Video. 3). To better control that tip bifurcation was a consequence of the physical interaction with the pillar, we quantified the dynamics of tip bifurcation by measuring the curvature of the pillar from the point at which the pillar touched the growing tip (time 0) every hour for 24 h. This allowed us to conclude that tip bifurcation was guided by pillars (Fig. 5e). Induced branching in hollow segments allows the preservation of the cytoskeletal apical polarity upon interaction with the pillars (Fig. 5f, Supplementary Fig. 7 and Supplementary Video. 4). Growth of branched bud tips continues beyond 3 days of culture (Supplementary Video. 5). Moreover, expression of the bud-tip cell marker Sox9 was down-regulated in correspondence with the bifurcation point that

was induced by the pillars during branching progression, with increasing Sox9 downregulation according to the cleft depth (Fig. 5g). Overall, we show that geometrical constraints can be created during the dynamic growth of lung rudiment 3D cultures to control lung tip bifurcation.

## Discussion

Here, we present a method that exploits near-infrared multiphoton laser irradiation of photosensitive hydrogels to build complex 3D structures around hydrogel-embedded live cells and organoids with micrometer accuracy, defined stiffness, and temporal control. With this technique it is possible to add an additional instructive environmental cue to evolving 3D biological structures. We show that, by shaping the 3D architecture of the environment surrounding organ-like organoids, we can regulate biological processes, such as size, shape, cell identity, migration and morphogenesis. The hydrogel-in-hydrogel structures fabricated with this method require diffusion of the photosensitive polymer within the pre-existing 3D biological construct. Thus, the approach proposed has broad applicability, allowing temporal and spatial design and guidance of 3D structures for targeting specific time windows and/or cell types in 3D cultures. The primary limitation of this bioprinting approach is the intrinsic limits of the optical properties of the microscope such as objective working distances and/or software set-up used.

The experiments on oSpC cultures showed that hydrogels can guide the orientation of neural axon sprouting in 3D space. Innervation plays a pivotal role as a driver of tissue and organ development as well as a means for their functional control and modulation[22,23]. During innervation, neural axons are guided by molecular, mechanical, and physical cues to reach their targets[24]. The understanding of neural axon guidance remains a challenge, for example during spinal cord regeneration. A major limit for modeling neural axon guidance in vitro so far has been the precise temporal regulation of guidance signals in the 3D environment[25]. Our experimental approach offers a unique ability to precisely control the sprouting of neural projections within the time and space of the culture. This opens new perspectives for deeper characterization and understanding of key players regulating the neural axon guidance in complex and dynamic 3D in vitro neural models.

Regarding our investigation of cancer organoid behavior, live bioprinting allowed assessment of differential cell migration in between pillars with different mechanical properties. During the stages of metastatic dissemination, cancer cells adapt their invasive and migratory strategies to the different 3D environments encountered[26–28]. Therefore, our approach could reveal how cancer cell migration is affected in 3D by different extracellular confinements. Moreover, we hypothesized that additive hydrogel-in-hydrogel bioprinting performed at multiple spatiotemporal points of the cancer organoid culture will allow investigation of dynamic hurdles posed by the invaded tissues, such as increased stiffness due to cancer associated fibroblasts or

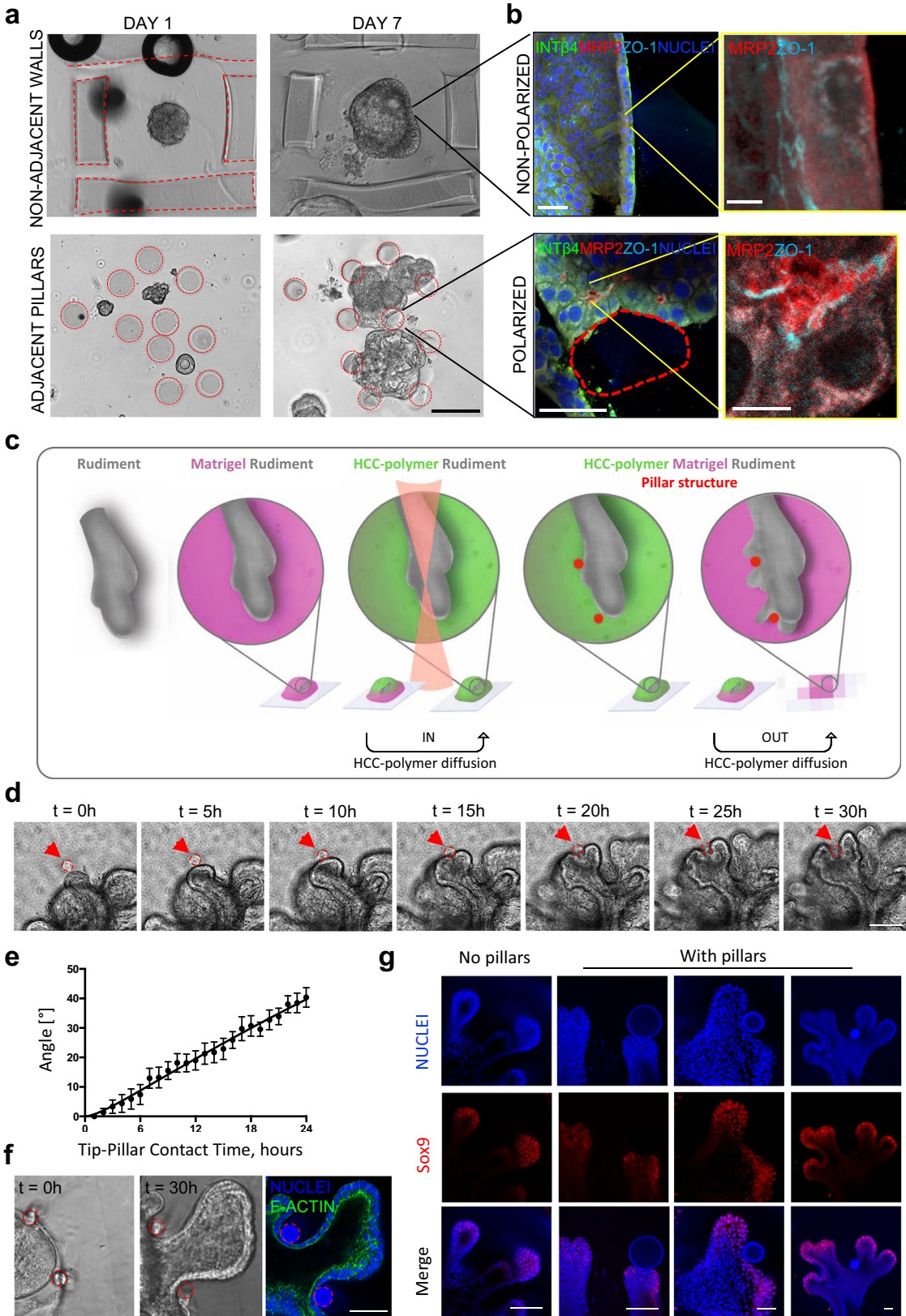

neoangiogenesis. This could be exploited, for example, to impose mechanical confinement on cells with low cell-cell cohesion and drive jamming[29] or to create tracks with different topology in the proximity of collectively migrating cells.

Supra-organoid physical constraints play a fundamental role in morphogenetic movements. Here, we used intestinal organoids to investigate the potential of our technique as the biological feature of the crypt/villus axis strictly relates to in vivo curvature angles, where it is known that apical constriction and bending gradient induce budding formation in the intestine[30,31]. We showed that our technology allows the design of structures that would permit the study of the curvatures that instruct small intestinal organoid budding.

**Fig. 5 | 3D geometrical constrains imposed by hydrogel-in-hydrogel live bio-printing on organoid and organotypic cultures. a, b** Induction of polarization in human fetal hepatocyte organoids. **a** Bright field images showing two printing strategies: distant walls not touching the growing organoid (above) and adjacent pillars touching the growing organoids (below) after 7 days of culture. Scale bar 100 μm. **b** Immunofluorescent panels showing non-polarized organoid distant from the printed structures (above) and a polarized organoid in correspondence of the printed pillars (below). Integrin beta-4 (INTβ4) shown in green, multidrug resistance-associated protein 2 (MRP2) shown in red, zonula occludens-1 (ZO-1) shown in cyan. Nuclei are stained with Hoechst (blue). Scale bars, 50 μm (large images, left) and 10 μm (higher magnification, right). **c–f** Ex vivo culture of mesenchyme-free lung epithelium rudiments, isolated from embryonic mice at stage E12.5. **c** Schematic of isolated fetal lung tips ex vivo culture and guided branching morphogenesis following 2 P bioprinting of gelatin pillars (red circles) in Matrigel. **d** 5-h interval snapshots of time-lapse reconstruction of budding lung tip during guided morphogenesis around pillar (red circle, indicated by red arrow). See full time-lapse in Supplementary Video. 3. Scale bar 150 μm. **e** Plot showing angle of tip bifurcation vs time from tip-pillar contact time (time 0) to 24 h of culture. Data are shown as mean ± s.e.m. of 8 independent replicates. **f** Bright-field images and immunofluorescent panel showing lung tip branching in between two pillars (red circles) and inner (luminal) polarity maintained (F-actin, green). Nuclei are stained with Hoechst (blue). Scale bar 100 μm. **g** Immunofluorescent panel showing lung tip branching in between pillar (red circle) with downregulation of sox9 (red) in correspondence of the pillar. Nuclei are stained with Hoechst (blue). Scale bars 50 μm.

A similar effect of a confinement-driven cell polarization was observed with liver organoids. Hepatocyte polarity requires proper compartmentalization of basolateral and apical proteins, as well as junction formation[32]. Recapitulation of correct polarization within organoids is essential to mimic organ function (i.e. mucus, acid, hormone, or bile acid secretion, depending on the tissue) in order to build a reliable and functional in vitro model. With our experiments we were able to demonstrate how a defined confinement by hydrogel-in-hydrogel bioprinting induces correct apical polarization within fetal hepatocyte organoids.

Finally, we investigated a more complex ex-vivo organ culture behavior, using lung epithelium rudiments. Lung branching morphogenesis shapes the respiratory tree through the iterative formation of branches from the surface of an epithelial bud tip and the subsequent branch outgrowth into the surrounding mesenchyme, according to a remarkably stereotyped and genetically hardwired process[33]. Here, we showed that ex vivo branching morphogenesis can be guided by forcing the bifurcation of bud tips through 3D-printed circular pillars. We confirmed the preservation of the cytoskeletal apical polarity, a prerequisite for branching initiation[34], as well as the loss of the bud tip identity represented by Sox9 downregulation[35] in tip cells undergoing bifurcation through the pillars. The extracellular environment is increasingly recognized as a key player in regulating the developmental processes of branching morphogenesis[36]. Our 3D model of guided branching provides a tool to investigate extracellular regulators of lung branching morphogenesis, such as interactions with extracellular matrix components and physical forces[37–39] that are increasingly recognized, yet poorly understood, to be integrated into the genetic patterns governing branching.

To conclude, the technique presented in this work allows simultaneously to (i) perform live imaging of organ-like 3D cultures, (ii) define the positioning and orientation of the desired hydrogel relative to a specific organoid with sub- or supra-organoid resolution, and tunable mechanical properties, (iii) perform single or additive hydrogel-in-hydrogel bioprinting at the desired time point of the organ-like 3D culture. Such technical features, combined with the biological effect of printed hydrogels, open the unique possibility to finely tune complex cellular responses in 3D organ-like structures to environmental cues over time. This hydrogel-in-hydrogel live bioprinting technology holds the potential to improve current in vitro organ modeling methods and to inspire innovative models for the investigation of dynamic and multicellular complex processes.

## Methods
### Hydrogel preparation
HCC-Gelatin, HCC-PEG and CMMC-PEG were synthesized as previously described[9]. Briefly, 7-hydroxycoumarin-3-carboxylic acid (HCCA) was conjugated to gelatin from porcine skin (Sigma-Aldrich). HCCA and 7-carboxymethoxy-4-methylcoumarin (CMMC) were conjugated to 10 kDa PEG amine tripentaerythritol core (4-arm PEG-NH2, Sigma-Aldrich) or 20kDA 8-arm PEG amine tripentaerythritol core (8arm-PEG-NH2, Sigma-Aldrich) to create biocompatible photosensitive compounds with alternate organic conjugated components (HCC-Gel, HCC-4-arm PEG, HCC-8-arm PEG, CMMC-4-arm PEG, CMMC-8-arm PEG) that crosslinks at defined wavelengths[9]. After synthesis, freeze dried powdered stock were combined with PBS at concentration of 10% w/v for HCC-Gelatin and 30% for HCC-4-arm PEG or HCC-8-arm PEG or CMMC-4-arm PEG or CMMC-8-arm PEG[9]. The resulting solutions were agitated at 800 rpm at 37 °C for PEG-based polymers or at 60 °C for Gelatin-based polymers until dissolving. The final solutions were kept at 4 °C, protected from light. HCC-Gelatin was reheated at 60°C for 10 min before use.

### Fluorescence Recovery After Photobleaching (FRAP) analysis
A drop of 3 μl of Matrigel® Basement Membrane Matrix Growth Factor Reduced (GFR, Corning, 354230) was cast on a glass coverslip (Bio Optica) equipped with custom-made polydimethylsiloxane (TESTA&C, D80DW184) well (3.5 mm diameter, 6 mm hight) and incubated at 37 °C for 4 min to allow MRF gelation. Subsequently, 15 μl solution of either fluorescein isothiocyanate (FITC)-dextran 40 kDa (2 mg/ml in PBS, (Sigma-Aldrich)) or FITC-dextran 500 kDa (2 mg/ml in PBS, (Sigma-Aldrich)) were loaded over the Matrigel droplet located within the PDMS well and incubated at 37 °C for 15 min.

For FRAP experiments we employed a Leica SP5 confocal microscope, equipped with an Argon laser 488 nm, 30 mW nominal power. The experiments were carried out with a 10x AIR objective, with 600 Hz scanning frequency at 90% of laser power (13% used for pre- and post-bleaching acquisition, 100% used for bleaching), and 10 frames of pre-bleaching acquisition, 20 frames of bleaching and 100 frames of post-bleaching acquisition (time between frames was set as 3 s). The bleaching areas were defined as follows: 30 μm in diameter circle, bleached with "Zoom-in mode", 100 μm bleaching z-plane. Fluorescence recovery was measured at the center of the bleached area consisting of a round ROI 30 μm in diameter. FRAP quantification is shown as the result of three independent experiments, with five measurements per each Matrigel droplet.

### Atomic force measurement
A liquid drop of HCC-Gelatin was added onto freshly prepared 100% Matrigel (Corning 354230) solid drops or onto 100% Matrigel drops prepared 2 days before treatment and maintained in DMEM/F12 (ThermoFisher Scientific) at 37 °C. Treated samples were incubated at 37 °C for 15 min. The volume ratio between photosensitive hydrogels and solid Matrigel drops was 2:1. Hydrogels were fabricate within the pre-existing Matrigel with drops by using Scientifica 2-Photon microscope according to our previous study[9]. For AFM measurements, Matrigel was depolymerized by using cold PBS-EDTA (0.5 mM EDTA, Invitrogen #AM9260G) under gentle agitation overnight at 4 °C as previously shown[9]. All the samples were analysed by using an Atomic Force Microscope, mounted on an Inverted Optical Microscope (XEBio, Park Systems, Korea) as previously reported[9].

## Organ-like cultures

**Intestinal and liver organoid cultures.** For mouse small intestinal organoids, LGR5-DTR-EGFP mice[16] were sacrificed by cervical dislocation and the intestine was harvested from the pylorus to the cecum, as previously reported[17]. The tissue was washed in ice-cold PBS (ThermoFisher Scientific), cleared of any mesenteric or fatty tissue and cut open longitudinally. A glass coverslip was used to remove the villi from the luminal mucosa and thoroughly washed in PBS. Tissue was further cut into 2–3 mm pieces and incubated in 2 mM ethylenediaminetetraacetic acid (EDTA−(Sigma-Aldrich) in PBS for 30 min at 4 °C. EDTA was removed, and tissue was vigorously shaken for 5 min in PBS with calcium-magnesium (ThermoFisher Scientific). The supernatant containing the crypts was collected and centrifuged at 800 rpm for 5 min at 4 °C (Hettich zentrifugen Rotina 420). The pellet was washed once with basal media Advanced DMEM/F12 media, supplemented with 1% of GlutaMAX, HEPES and Penicillin/Streptomycin (ADMEM + / + / + , all ThermoFisher Scientific) and centrifuged at 1000 rpm. The pellet was re-suspended in Matrigel® Basement Membrane Matrix Growth Factor Reduced (GFR) (Corning 354230) and plated onto a 24-well plate (Corning). Primocin 1X (ThermoFisher Scientific) and 10 μM ROCK inhibitor (Tocris 1254) were added after isolation. All the reagents supplemented for the mouse small intestinal organoid cultures are reported in Supplementary Table 3. For human fetal hepatocyte organoids, human fetal livers were collected via the Joint MRC/Wellcome Trust Human Developmental Biology Resource under informed ethical consent with Research Tissue Bank ethical approval (18/LO/0822, project 200525). One eight post conception week (PCW) liver was washed in PBS, cut into 1–2 mm cubes and digested for 10 min in HBSS (ThermoFisher Scientific) + EDTA 2 mM at 37 °C. HBSS was discarded, and liver cubes were further dissociated in collagenase type 1 (Sigma-Aldrich) for 10 min at 37 °C. Liver cubes were pipetted thoroughly to facilitate dissociation until a single cell solution was obtained. Cells were pelleted at 800 rpm for 5 min at 4 °C and washed twice in ADMEM + / + / + . Finally, cell pellet was resuspended in Matrigel and plated in 24-well plates (Corning). Primocin 1X (ThermoFisher Scientific) and 10 μM ROCK inhibitor (Tocris 1254) were added after isolation. Liver organoids were cultured following the protocol previously published[10]. Cells were passaged every 6–8 days. All the reagents supplemented for the human liver organoid cultures are reported in Supplementary Table 4. To passage the organoids, Matrigel droplets were thoroughly disrupted by pipetting in the well and transferred to tubes in ice. Cells were washed with 10 mL of ice-cold ADMEM + / + / + and spin at 200 g at 4 °C. For single cell disaggregation, organoid pellets were incubated in TrypLE Express (ThermoFisher Scientific) for 5 min at 37 °C. Organoids were thoroughly pipetted to help disaggregation, then single cells were spin at 200 g at 4 °C and plated in Matrigel droplets. Following single cell dissociation, ROCK inhibitor 10 μm was added to the culture medium. For mechanical disaggregation into cell clumps, the pellet was resuspended in 1 mL of cold ADMEM + / + / + and organoids were manually disrupted by narrow (flamed) glass pipette pre-wet in BSA 1% (Sigma-Aldrich) in PBS, to avoid adhesion to the glass. Cells were washed, pelleted and supernatant was discarded. Almost-dry pellets of disaggregated organoids were thoroughly resuspended in cold liquid Matrigel, aliquot in 30-40 μL droplets in petri dishes (Corning), and incubated at 37 °C for 30 min to form a gel.

**Cancer organoid cultures.** A549 lung adenocarcinoma cells (ATCC) were cultured in RPMI 1640 medium (Sigma-Aldrich) supplemented with 10% of fetal bovine serum (FBS, ThermoFisher Scientific) and 1% of Penicillin/Streptomycin (ThermoFisher Scientific). For organoid generation, mycoplasma-free A549 cells were seeded in 90% Matrigel (Corning) at a final concentration of $0.3 \times 10^6$ cells mL$^{-1}$. The cells embedded in Matrigel (25 μL each) were transferred into a 24-well culture plate (Corning) and incubated at 37 °C in 5% $CO_2$ with medium change every 2–3 days. After 7 days, Matrigel was dissolved by using Cell Recovery Solution (Corning) for 20 min on ice. The organoids were gently resuspended in culture medium, and re-embedded in collagen hydrogel at a density around 40 organoids per hydrogel. To prepare the collagen hydrogel, organoids were mixed with type I collagen (rat tail derived, ThermoFisher Scientific) solution and adjusted to neutral pH by NaOH (1 N, Sigma-Aldrich) to form a physical cross-linked hydrogel with a final concentration of 2.4% (w/v) after incubating at 37 °C for 20 min. The organoids in hydrogels (12 μL each) were cultured at least 24 h before printing.

**Mouse fetal lung organotypic cultures.** All animal experiments were performed by personnel having UK Home Office Personal Licence (PIL I7ED92582) in line with ethical approval. Wild type CD-1 mice were mated and marked as E0.5 pregnant when they presented a vaginal plug. Pregnant mice were euthanized by cervical dislocation at E12.5, corresponding to pseudoglandular stage of lung development, and lung epithelium rudiments were cultured as previously reported[21]. Briefly, embryos (obtained through time-mated pregnancies) were extracted, and lungs harvested under a stereomicroscope. Lungs were isolated from the embryos and washed in PBS (ThermoFisher Scientific) and subsequently treated with 8 U/mL Dispase (ThermoFisher Scientific) for 2 min at room temperature. Mesenchymal tissue was mechanically removed with tungsten needles (Interfocus) and distal tissue including at least one bud tip was isolated and embedded in a 5 μL Matrigel drop to allow for 3D culture on sterile coverslips. After Matrigel gelation, 250 μL of DMEM/F12 supplemented with 1% Penicillin/Streptomycin, 0.1% BSA (Sigma-Aldrich), 1% Insulin-Transferrin-Selenium (ITS) (all ThermoFisher Scientific), 200 ng/mL human Recombinant FGF10 (Peprotech) and 1 μM CHIR99021 (Tocris). All the reagents supplemented for the mouse fetal lung organotypic cultures are reported in Supplementary Table 5.

**Rat fetal spinal cord organotypic cultures.** For rat fetal spinal cord cultures, E14 fetuses were obtained from pregnant Sprague Dawley rats (Charles River Laboratories, Wilmington, MA, USA). Rat fetal spinal cords were isolated following the ethical approval (D2784.N.I6Q) and culture as previously described[12]. Briefly, isolated spinal cords were cut in three sections and cultured within 15 μL drop of 100% Matrigel (Corning) casted onto glass coverslips. Samples were cultured in Neurobasal medium (Gibco 21-103-049), B-27 supplement (Gibco 17-504-044) 1X, 2% Horse serum (Gibco 16-050-122), 0.5 mM GlutaMAX Supplement (Gibco 35-050-038), 25 μM 2-Mercaptoethanol (Gibco 31-350-010), 25 μM L-Glutamic acid (Sigma-Aldrich, G5889), Gentamicin/Amphotericin (Gibco R01510), 10 ng/mL ciliary neurotrophic factor (CNTF, PeproTech 450-13), and 10 ng/mL glial-cell-line-derived neurotrophic factor (GDNF, Peprotech 450-10).

## Hydrogel-in-hydrogel bioprinting

Organoid or organotypic culture pre-seeded into droplets of hydrogel (Matrigel Corning or collagen, ThermoFisher Scientific) were cultured onto 13 mm coverslips (VWR International) in 24-well plates (Corning). On the day of the print, coverslips were removed with sterile tweezers, dried with sterile paper and incubated with photosensitive-polymers. For photosensitive hydrogel diffusion, a liquid drop of HCC-PEG or HCC-Gelatin was added onto the 3D culture and incubated at 37 °C for 15 min (Matrigel-based cell cultures) or 4 h (Collagen-based cultures). The volume ratio between photosensitive hydrogels and solid hydrogel drops was 2:1. The multiphoton microscopes used for 2P-mediated hydrogel crosslinking were Scientifica 2-Photon microscope, Zeiss Examiner.Z1 Multiphoton LSM880 Confocal microscope equipped with Solent Scientific Incubator Chamber and Multiphoton Leica SP8 confocal microscope. After the bioprinting, the coverslip was moved into another 6 mm petri dish (Corning) filled with 10 ml of basal media and incubated at 37 °C for 5 min to remove the

un-crosslinked photosensitive-polymer. Finally, the coverslip was put in a 24 multi-well plate (Corning) with 500 µl of organoid or organotypic culture media.

## Multiphoton crosslinking settings

For experiments performed with murine small intestine and human fetal liver organoids, and murine lung organotypic cultures, we used Zeiss Examiner.Z1 Multiphoton LSM880 Confocal microscope. Hydrogel design for primordial small intestine was performed using AutoCAD 2020. Optimized settings for crosslinking in murine small intestine and human fetal liver organoid samples were as follows: pixel resolution 1640 × 1640, pixel dwell time 0.98µs, Zoom 1.2X, laser wavelength 800 nm, stack depth 2.57 µm, laser power 67% for Gelatin-based hydrogels and 40% for PEG-based hydrogels, 10X immersion objective. Optimized settings for hydrogel crosslinking in murine lung organotypic culture samples were as follows: pixel resolution 1640 × 1640, pixel dwell time 0.98µs, Zoom 2X, laser wavelength 800 nm, stack depth 1 µm, laser power 40% for PEG-based hydrogels, 4X immersion objective. Hydrogels were crosslinked by imaging designed ROIs generated by the Zeiss software. Z-stack depth was specifically defined on the basis of 2P-live imaging and according to the organoid size. For experiments performed with cancer organoids, we used the Multiphoton Leica SP8 confocal microscope. Pillar structures were printed around the organoids according with the design by using the multiple ROIs generated by the Leica software. For experiments performed with spinal cord cultures, Scientifica 2-Photon microscope. Hydrogels were fabricated according to our previous study[9].

## Immunofluorescence and imaging acquisition

Matrigel droplets with embedded photo-printed organoids were fixed in 0.5% glutaraldehyde (Sigma-Aldrich) dissolved in PBS with Ca$^{++}$/Mg$^{++}$ for 15 min at RT, and then extensively washed in PBS. Free aldehydes were quenched with an incubation of 0.1 M NH$_4$Cl (Sigma-Aldrich) for 1 h. Whole mount staining was performed by blocking and permeabilizing the cells with PBS-Triton 0.5% (Sigma-Aldrich) with BSA 1% (Sigma-Aldrich) for 2 h at room temperature (RT). Primary antibodies were incubated in blocking buffer for 48 h at 4 °C in rotation, and then extensively washed in PBS-Triton 0.5% at RT. Secondary antibodies were incubated overnight at 4 °C in rotation and extensively washed. The list of the used antibodies is reported in Supplementary Table 6. Samples were mounted in TDX mounting medium (Sigma-Aldrich). Bright field images and EGFP fluorescence images were acquired at a Leica DMIL microscope and DFC420C camera. Immunofluorescence and live imaging were acquired by using Zeiss LSM 710, Zeiss Examiner.Z1 Multiphoton LSM880 Confocal or Multiphoton Leica SP8 confocal microscopes.

## Cell viability assay

For cell viability assays, LIVE/DEAD™ Viability/Cytotoxicity Kit, for mammalian cells (Thermo Fisher L3224), was used following supplier instructions. Briefly, organ-like cultures were washed with the specific basal cell culture medium and incubated in basal medium with calcein-AM (3 µM) and ethidium homodimer-1 (3 µM). Cells in Matrigel droplets were washed twice and analyzed by fluorescence imaging. In oSpC 3D cultures, viability was investigated 4 h after hydrogel-in-hydrogel live bioprinting. For cancer organoid 3D culture, cells were analyzed 1 day after first or second round of bioprinting. For mSIO experiments, samples were analyzed at day 3 and day 7 post printing.

## Imaging preparation and analysis

We used ImageJ software for adjustments of levels and contrast, maximum and standard deviation intensity projections and 3D reconstructions. All the quantification reported have been performed with ImageJ software. Quantification of hydrogel swelling was performed by measuring the area of the hydrogels. For directionality analysis, directionality ImageJ plugin was used to analyze bright field and TUJ-1 immunofluorescence images of the neural projections departing from the central body of the spinal cord within the volume the Matrigel, in presence or absences of HCC-Gelatin hydrogels. The measurements were performed as previously described[12]. Images with completely isotropic content are expected to give a flat histogram, whereas images in which there is a preferred orientation are expected to give a histogram with a peak at that orientation. The quantification was expressed as the mean of 4–5 independent biological replicates. For lung tip bifurcation, we calculated the right and left angles generated between the tangent to the pillar and the parallel direction of tip bifurcation by using angle tool plugin of ImageJ software. Mean angles for each sample were plotted against time.

## Statistical analysis

All analyses were performed with GraphPad prism 6. We expressed data as mean ± s.e.m or mean ± s.d of multiple biological replicates (as indicated in the figure legends). We determined statistical significance by unequal variance Student's $t$-test, one-way analysis of variance (ANOVA) and Tukey's multiple comparison test or two-way ANOVA and Sidak's multiple comparisons test. $P$-value <0.05 was considered statistically significant.

## Reporting summary

Further information on research design is available in the Nature Portfolio Reporting Summary linked to this article.

# Data availability

All relevant data supporting the key findings of this study, as well as the raw image data generated in this study, are available within the paper and its Supplementary Information or from the corresponding author upon reasonable request. Source data are provided with this paper.

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

## Acknowledgements

This work was supported by 2017 STARS-WiC grant of University of Padova, Progetti di Eccellenza CaRiPaRo, TWINING of University of Padova, Oak Foundation Award (grant no. W1095/OCAY-14-191), AFM Telethon (grant code 23284), 'Consorzio per la Ricerca Sanitaria' (CORIS) of the Veneto Region, Italy (LifeLab Program), European Research Council (grant agreement 101055300 - ReprOids) to N.E., and the STARS Starting Grant 2017 of University of Padova (grant code LS3-19613) and IRP Consolidator Grant (grant code: 21/05 Irp) to A.U.. We thank Prof. Frederic de Sauvage for providing Lgr5-DTR-GFP mice. G.G.G. was supported by the NIHR Great Ormond Street Hospital Biomedical Research Centre and the BRC Catalyst Fellowship. P.D.C. and N.E. were supported by the Oak award W1095/OCAY-14- 191. P.D.C. is supported by NIHR Professorship and the GOSH Children's Charity. F.M. is supported by a NIHR BRC Catalyst Fellowship and a Rosetrees Trust grant (grant code Seedcorn2020/100052). P.D.C. and F.M. are supported by Building Respiratory Epithelium and Tissue for Health (BREATH) Consortium for Lung Regeneration (LongFond, Dutch). S.S. is supported by Japan society for the promotion of science overseas research fellowships (310072). G.L.G. is supported by the Wellcome Trust (211112/Z/18/Z and 211112/Z/18/A). Y.D. is supported by Science and Technology Commission of Shanghai Municipality, 21140901500. M.Mo. is supported by the STARS Consolidator Grant 2021 from University of Padova and MFAG 2021 (#25745) from the Italian Association for Cancer Research (AIRC). All research at Great Ormond Street Hospital NHS Foundation Trust and University College London GOSICH, Zayed Centre for Research into Rare Disease in Children is made possible by the NIHR Great Ormond Street Hospital Biomedical Research Centre. The views expressed are those of the author(s) and not necessarily those of the National Health Service, the NIHR or the Department of Health. We would like to thank the Human Developmental Biology Resource (HDBR). We thank David Martos Ruiz for creating Figs. 1a and 5c.

## Author contributions

A.U., G.G.G., Y.D. and N.E. designed the study. G.G.G. derived and characterized liver and intestine organoid cultures. L.B. and P.R. contributed to hydrogel-in-hydrogel printing characterization experiments. O.G., M.Ma. and G.S. contributed to the organoid printing experiments. F.M. design the experiment with fetal lung epithelium. F.M. and S.S. performed the experiments with the fetal lung epithelium. D.S. contributed to experiments with the fetal lung epithelium. A.U., P.R. and V.S. performed the experiments with organotypic spinal cord cultures. P.C. helped with the derivation of organotypic spinal cord. Y.D., X.W. and J.Q. performed the experiments with cancer organoids. M.G. characterized samples with AFM. A.U. and N.E. supervised the project. M.Mo., H.C., M.N., P.D.C., G.L.G. helped to design the experiments and critically discussed the manuscript. A.U., G.G.G. and N.E. wrote the manuscript.

## Competing interests

N.E. and O.G. have an equity stake in ONYEL Biotech s.r.l. A.U. and N.E. are inventors of a patent for the use of HCC- and CMM-hydrogels (patent applicant: ONYEL Biotech s.r.l.; patent number EP4138941). All other authors have no competing interests.
