## [Peer Review File · Nature Communications]

Reviewers' comments:

Reviewer #1 (Remarks to the Author):

In this manuscript, the authors demonstrated an interesting approach and build structural elements with the use of 2P crosslinking while culturing organoids, which was named as "4D Bioprinting." 4D Bioprinting/printing has been shown for substantial programmable shape changes after (bio)printing and 3D printing of structural pillar for guiding organoid growth is different than what has been shown in other 4D (bio)printing efforts; however, the authors already demonstrated 2P crosslinking of structural elements confining organoid growth in their Nature Biomed. Eng. Paper last year. "In situ printing of structural guiding elements" or something similar could be a better term than 4D bioprinting as the authors did not print any biologics at all and printing at different time points is not truly 4D. In addition, the use of 4D bioprinting has not been well demonstrated in biological examples as most of the examples did not utilize 4D bioprinting. The rationale for printing structural elements at different time points has not been well justified with the given examples.

Major comments:

1. The overall process in this study can be considered as 3D printing at multiple time points. Generally, 4D printing refers to a dynamic structure with a programmed characteristic that transforms into a desired structure by estimated internal/external stimuli.
2. The organoid alignment and the interactions between an organoid and reconstructed hydrogel were presented in their previous study. Examples are new but the novelty in the concept presented in Figures 3 and 4 is limited.
3. Is printing pillars at different time points safe for cells? How can organoids survive long time in uncrosslinked hydrogels?
4. How did the authors determine at which time points they should print a pillar around an organoid? Organoid growth possess uncertainties and it is not clear if previously printed pillars will yield the same outcome.
5. What is reproducibility of guiding cell morphology (i.e. branching initiation or bifurcation)?
6. For printing pillars, printing is limited to vertical direction only around an organoid. It will likely that organoids will grow in upward and downward directions more if they are confined from sides, which will bring further uncertainties. Could the authors address this limitation or relevant discussion should be made for future work?
7. To support the presented concept, more profound examples should be demonstrated. With the given single example in Fig 2, it is not clear why someone needs to perform 3D printing of pillars at different time points in situ. It is not clear if structural elements in Examples in Figures 3 and 4 were printed at different time points; rather, they seem to be printed at time 0.
8. What would be difference between the use of a pre-designed 3D-printed structure versus a 4D-printed structure at different time points? Indeed, 3D printing of the scaffold prior to cell seeding could enable more advanced confined structures as top and bottom surfaces can also be crosslinked.
9. Bifurcation by pillars is interesting but limitations should be discussed as the bronchial segments should be hollow.

Minor comments:

1. Revise the typo "fist" on Page 5.
2. The scale bars are randomly present, so please make them consistent for the figures.
3. Figure 4C is confusing at first look. Colors can be represented in a legend.

Reviewer #2 (Remarks to the Author):

Reviewer Comments to Author

This paper conveys a fabrication method to guide the morphogenesis of cells in vitro by imposing geometrical cues inside hydrogel using photo-sensitive polymers and two-photon mediated printing.

This research is distinguished from the current printing methods where fabricated structures are confined to initial designs. Furthermore, this paper presents the versatility of the developed live printing technique, especially applications in guiding the branch-like structure of the intestine and respiratory tree. This paper will give a lot of inspiration for future researchers in this field, therefore can be recommended for publication with some major comments for authors to carefully consider.

Major comments

1. For Figure 2c-f, the reviewer is wondering what kind of dynamic hurdles the authors especially intended to mimic, and what kind of adaptation the cancer cells showed exactly when encountering the bioprinted environment. Please add some clear explanations and references if necessary.
2. In this manuscript, the most extended observation timepoint after bioprinting is day 14. However, morphogenesis processes in vivo generally take a longer time than that, so the reviewer recommends the authors provide long-term (i.e., 4 weeks) results of the culture.
3. Basically, it is more logical to give the readers a quantified accordance rate between the intended design and the bioprinted structure to demonstrate that the 3D structures were faithfully reproduced as designed, for example, in Figure 3a.
4. In Figure 3d and Supplementary S5d, the authors presented the quantification data of the central and branched area covered by mSIOs. Please define what SIO1 to SIO6 means in the main manuscript.
5. The reviewer thinks the readers would expect to see how the mSIOs adapted to both circular central parts and the branched sides. Please consider presenting Supplementary Figure S5b and S5d in the main figure, for example, following Figure 3c-d.
6. Continuing on comment 5, please give some explanation about why the area occupied by the mSIOs are largely different from SIO1 to SIO6 (especially in the branch) and why the plots are not continuous through day 0 to 9 in Figure S5d. It would be great if the authors can present full images of Figure 3b or Figure S5b to show the readers that the adaptation of mSIOs occurred in every branch.
7. The reviewer would like to kindly ask the authors to state how long does it take to print the 3D structures in the manuscript.

Minor comments

1. Please carefully check if there is Figure 1i (page 4, line 18).
2. Figure 1 legend: i, ii, iii are not labeled in the figure, and please check if the explanation of legend e) is appropriate. There is no scale bar in f.
3. Suppl. Fig. S1a: The blue and green letters are difficult to read.
4. Please correct the typo (stare→started) on page 5, line 24.
5. Figure 2 legend: There is an unnecessary bracket in the legend of d.
6. Suppl. Fig. S4a: Please add arrows to indicate the pillars.
7. Figure 3 legend: Please carefully check if the Suppl. Table 1 and 2 are correctly cited. The reviewer would recommend adding an appropriate, detailed statistical analysis part in the supplementary document if needed. It seems the legend d is not a complete sentence. Please carefully revise it.

8. Suppl. Fig. S5e: It is difficult to read the blue letters.

9. F-actin (ACT) is not used in Fig. 4b. Please carefully correct the manuscript (page 7, line 24).

10. Please clearly explain what kind of lung epithelial cell was used in Figure 4d-f in the main manuscript.

11. The reviewer thinks there's a redundancy in the paragraph starting with 'Then, we investigated the possible biological effect of live bioprinting' on page 8. Please concise this part or rephrase.

Reviewer #3 (Remarks to the Author):

The authors present a novel and interesting way to use 2P crosslinking to allow for the printing of 3D structures within pre-existing hydrogel-based matrices (i.e. Matrigel) through the diffusion of the photo-sensitive polymer in the pre-existing hydrogel structure. This allows for alterations over time within the 3D culture environment to steer and/or influence the development of organoid cultures. The methodology appears to be sound and the use of a range of different cell types/ organoids is a strong aspect of the manuscript.

The presented concept is derived from the previous work of the authors (reference 9 in the manuscript). In my view, the main novelty of the present work is that the photo sensitive polymer is not mixed with the Matrigel, but, after seeding of the cells, allowed to diffuse in the cell-laden structures. This could then allow modifications in, e.g. mechanical, properties also later to be added to the hydrogel structures. This novelty is well visualized in Figure 1A. Subsequently, number of potential applications of the technology are explored using different "organ-like" and organoid structures.

While the concept is very interesting and will provide an additional tool in the field of bioprinting, the technological step is somewhat incremental from the authors previous work (especially regarding reference 9). The provided examples are "very nice", but not "overwhelming". In view of this, I do feel the work should be published, but probably in a more specialized journal

Some more minor comments and suggestions

1. In the Abstract "2P" is not explained

2. Use of "4D" in the field is not very defined. The use of "live 4D" does also not really add. I would suggest to use the term "spatio-temporal" as this is more descriptive.

3. What do the authors mean when using "organ-like"?

4. The meaning of the following sentence is unclear (consider rephrasing): "Newly fabricated hydrogels can be tailored to the specific culture system requests and at the desired culture time, matching the spatiotemporal requirement to control 3D organ-like cultures. "

5. How does the presence of the organoids/cells influence the resolution?

6. Did the authors also evaluate the mechanical characteristics of the newly formed structures (beyond Fig 1h)? This could really provide this insights on how this technology could mimic the mechanically changing micro environment.

7. The intestinal organoids were only followed for a short period of time. Probably the lumen would quickly be overgrown due to the lack of a mechanism to remove the dislodged cells (see also work of Lutolf Lab). Have the authors considered to implement this into a (micro) perfusable system?

8. Regarding the budding of the lung organoids, while budding it self happens spontaneously (See also fig 4d); what is the main advantage of being able to control this? Do the buds at the later stages of culture fuse again (when they have grown around the pillars)?

Here we present a detailed point-by-point rebuttal to the issues raised by the reviewers.

Reviewer #1 (Remarks to the Author):

In this manuscript, the authors demonstrated an interesting approach and build structural elements with the use of 2P crosslinking while culturing organoids, which was named as “4D Bioprinting.” 4D Bioprinting/printing has been shown for substantial programmable shape changes after (bio)printing and 3D printing of structural pillar for guiding organoid growth is different than what has been shown in other 4D (bio)printing efforts; however, the authors already demonstrated 2P crosslinking of structural elements confining organoid growth in their Nature Biomed. Eng. Paper last year. “In situ printing of structural guiding elements” or something similar could be a better term than 4D bioprinting as the authors did not print any biologics at all and printing at different time points is not truly 4D. In addition, the use of 4D bioprinting has not been well demonstrated in biological examples as most of the examples did not utilize 4D bioprinting. The rationale for printing structural elements at different time points has not been well justified with the given examples.

We would like to thank the Reviewer for the detailed comments on our manuscript that we used as a base to improve our work. First, we acknowledge “4D bioprinting” could be interpreted differently by scientists in other fields and have now avoided this phrase throughout the text. In the current version of the manuscript, we decided to change the title to “Live bioprinting for structural cell guidance and dynamic control of organoid and organotypic cultures” following the Reviewer’s suggestion.

What we demonstrate, including with new experiments, is how spatiotemporal control of cultures can be reliably achieved using the technology we describe. We previously referred to 4D since we were able to perform such cellular control within the time of culture and in 3D space. Importantly, there is a main difference between the current work we are presenting and what was shown in the Nature Biomedical Engineering paper: we were previously only able to control the initial geometric constraints of cultures when first seeded but have now developed the technology to the point of being able to change geometry at any time during culture including after differentiation of specialized cell types. In this manuscript we show that the photosensitive hydrogels in their liquid form can be loaded at any desired time point of the 3D organ-like cultures, which are already embedded in solid Matrigel/Collagen. This is completely unexplored and new features, which require a combination of a proper diffusive properties as well as high cross-linking efficiency of photo-sensitive polymers within a pre-existing 3D gel. The fact that we are demonstrating the realization of these two aspects, is in our opinion surprising. As consequences, these aspects

allows bioprinting of the hydrogels at any desired time point, according to the dynamic change of 3D organ-like cultures.

We must point out the novelty and technological advancement of the work described based on the spatiotemporal versatility of the bioprinting that can be tailored on the specific requirements of the 3D organ-like culture (in terms of time, space, material, and mechanical properties).

Major comments:

1. The overall process in this study can be considered as 3D printing at multiple time points. Generally, 4D printing refers to a dynamic structure with a programmed characteristic that transforms into a desired structure by estimated internal/external stimuli.

Following the Reviewer's suggestions, we eliminated the 4D definition throughout the entire text and we define it as hydrogel-in-hydrogel live bioprinting. This is considered as 3D printing at multiple time points, that has not been described before. We have now better underlined that the described strategy is 3D printing within the same dynamically evolving 3D cultures. The 4D was not referred to the printed object, but to the idea of controlling the 3D culture in space and time (at multiple time points).

2. The organoid alignment and the interactions between an organoid and reconstructed hydrogel were presented in their previous study. Examples are new but the novelty in the concept presented in Figures 3 and 4 is limited.

In our previous study we showed that the photosensitive hydrogels were premixed with Matrigel, both in their liquid form. This allowed the printing of the structures only at the initial time point of the cell seeding. Moreover, these structures were tentatively positioned in a 3D domain that the operator considered to be relevant after a certain time of organoid culture. This also depends on the stochasticity of in vitro morphogenesis and cannot be fully predictable.

In the current version of the manuscript, we are presenting the idea of printing instructive structures at a specific time point and in a specific 3D domain, based on the ongoing morphogenesis of the culture, including after the emergence of specialized cell types such as neurons. We show how a photosensitive hydrogel can be loaded at the desired time point of the culture, opening the possibility to adapt the design of the guiding elements to

the morphogenesis of the organoid/organotypic culture. This gives the operator the possibility to decide upon a certain time of 3D culture, which organoid and/or which part of it will be subjected to guidance according to the experimental requirement and to the final aim of the analysis. Our approach also allows to position and orientate the structure relative to the organoid with a micrometer accuracy resolution. This novel concept has been only theorized and described as an essential necessity in the field of organoid, but never reported before this work.

Regarding the novelty of the unrevised **Fig. 3** and **4** (now **Fig. 4** and **5**), **Figure 3** showed how supra-organoid structures can guide mSIO morphogenesis to model the development of primordial small intestine in vitro. Such complexity in 3D printing was never achieved before with this level of detail. Unrevised **Figure 4** was showing the effect of hydrogel in polarizing liver organoids and guiding lung rudiments bifurcation. None of the two presented experiments were previously reported.

3. Is printing pillars at different time points safe for cells? How can organoids survive long time in uncrosslinked hydrogels?

We would like to clarify that the polymeric solution once equilibrated in the solid Matrigel/gelatin, are printed with the multiphoton irradiation and the uncross-linked hydrogel is washed away shortly after the procedure. We better clarified this in the material and methods section and in the new figure and text of the manuscript.

Nonetheless, to address the Reviewer's doubts regarding safety and cytocompatibility, we performed new experiments on different organ like structures and evaluated cell viability after the printing at specific time points. We present in **new Fig. 2b-d** (spinal cord), **new Suppl. Fig. 3e** (cancer organoids), and **new Suppl. Fig. 5b** (mSIOs) the absence of cell death compared to unprinted controls, as reported below for the Reviewer's convenience.

New Fig. 2b-d

New Suppl. Fig. 3e

New Suppl. Fig. 5b

4. How did the authors determine at which time points they should print a pillar around an organoid? Organoid growth possess uncertainties and it is not clear if previously printed pillars will yield the same outcome.

The Reviewer correctly highlights the strength of the method we are presenting. This question is strongly related to the novelty of the presented work and the need of this technology in the field. Indeed, the possibility to define when the guiding entities could be printed within the ongoing culture helped us to increase the reproducibility of our experiments. The exact timing was determined independently for each experiment. For example, the printing performed for organotypic spinal cord cultures was defined by the axon sprouting event (**new Fig. 2a-c**). The double printing with cancer organoid was performed when the first set of printed pillars was in contact with the growing organoid (**new Fig. 3f-h**). The printing performed in liver or intestinal organoid cultures was defined by morphological similarity (in terms of organoid size and cell organization) of the investigated organoids in the different experiments performed. In regards of lung rudiment experiments, the printing was performed according to the morphology of the culture to guide tip bifurcation.

New Fig. 2a-c

New Fig. 3f-h

Overall, the presented approach allowed us to temporally follow the dynamic morphogenesis of the organoid in vitro and to perturb it at similar conditions across

experiments. Moreover, this approach allows to guide morphogenesis of only the organoids that are currently growing in the culture, avoiding the unpredictable non-growing organoids, thus limiting the waste of time and costs caused by unsuccessful experiments.

5. What is reproducibility of guiding cell morphology (i.e. branching initiation or bifurcation)?

In general, we have observed highly reproducible results in the experiments that we performed. For instance, regarding guidance of neural axon sprouting, hydrogels always showed the ability to promote neural axon alignment in every experiment performed, as shown in **new Fig. 2 e-f** (below). Regarding lung experiments, the reproducibility of guiding the branching is very much dependent on the orientation of the tip. Considering that the two-photon microscopes we have available can only print in one direction, i.e. perpendicular to the glass slide supporting the Matrigel drop, the “guided” branching can be achieved only if the direction of the expanding tip and adjacent stalk is perpendicular to the pillar. In this scenario the reproducibility is about 50%.

New Fig. 2 e-f

6. For printing pillars, printing is limited to vertical direction only around an organoid. It will likely that organoids will grow in upward and downward directions more if they are confined from sides, which will bring further uncertainties. Could the authors address this limitation or relevant discussion should be made for future work?

We acknowledge the Reviewer’s doubt, and we performed new experiments to address this issue. In the previous version of the manuscript, we projected a defined ROI along the Z axis around organoid for sake of simplicity. However, the hydrogel crosslinking can be obtained along any line scan. So, there are no restriction as any line-scan trajectory can be performed according to the design of the software.

We discuss this aspect in the manuscript, highlighting that the major limitation is given by the software that control the light trajectory and the optical setup. We have now included in the manuscript new experiments to better clarify this point (**New Fig. 2**, attached below) and included new text in which we address this issue. Here we showed that the printing can be achieved according to ROI or line-scan and at different Z position, and not limited to the lateral surrounding of an organoid. In **new figure 2a** we showed that hydrogel can be fabricated below the central body of oSpCs, thanks to the ability of the pulsed laser to pass through the cells and focalize the voxel below them. The accuracy of the printing was also confirmed when hydrogels of different sizes were fabricated at different Z axis and same XY axis (**new Figure 2b-c**). The printing does not impinge cell viability, as shown by calcein staining and integrity of embedded neural axons presented in **new figure 2a-d**. Finally, we have now also shown that line-scan printing can be safely performed in the presence of cells according to the desired design of the object (**new Fig. 2e-f**).

New Fig. 2

7. To support the presented concept, more profound examples should be demonstrated. With the given single example in Fig 2, it is not clear why someone needs to perform 3D printing of pillars at different time points in situ. It is not clear if structural elements in Examples in Figures 3 and 4 were printed at different time points; rather, they seem to be printed at time 0.

As discussed before, there is a real gain in the possibility to print hydrogels at the desired time point of organoid/organotypic culture. We are presenting new data in **new Figure 3** where we show how we can print at subsequent time points in the same cell culture. We understand the point raised by the reviewer, and in the new manuscript we have better discuss this and we clarified the points. The elements presented in **new Fig. 3f** are printed at D1 of culture post-organoid seeding. Once the cancer organoid grows enough to fill the space imposed with the first set of pillars (at Day 6 in our experiments) we were able to print a second set of pillars surrounding the first “cage”, **new Fig. 3g**. The organoids can then invade the second set of pillars during the following days of culture, **new Fig. 3h**.

New Fig. 3

8. What would be difference between the use of a pre-designed 3D-printed structure versus a 4D-printed structure at different time points? Indeed, 3D printing of the scaffold prior to cell seeding could enable more advanced confined structures as top and bottom surfaces can also be crosslinked.

We wish to clarify that we can print also on top and on bottom of the organoid embedded within Matrigel. Pre-printing of scaffolds would require prior knowledge of the shape, rate of expansion and individual future morphology of organoids at the time of

seeding. Imposing geometric constraints and differential material properties before cellular commitment also has the potential to alter differentiation trajectories. All these limitations are avoided by printing after the organoid has matured to the desired starting point.

9. Bifurcation by pillars is interesting but limitations should be discussed as the bronchial segments should be hollow.

We understand the reviewer's doubts. We have not observed any significant difference in the epithelium structure and in the preservation of hollow bronchial segments in the presence or absence of 3D-printed pillars. The 3D-printed pillars functioned as spatial constraints that guided the bifurcation. To better highlight this, we have included an additional Supplementary Figure and Video, i.e., **new Suppl. Fig. S7** and **new Suppl. Video S4**, which show the 70 μm Z-stack of an immunofluorescence staining of a representative lung epithelium sample that underwent branching in the presence of two pillars (see **Fig. 5e** for reference) and highlighted the empty areas in the inner part of the tissue. Overall, this images sequence shows that the stalk regions of the epithelium, i.e. the regions that give rise to the bronchial segments of the airways, generated in the presence of the guiding 3D-printed structures, are hollow (image below).

New Suppl. Fig. S7

Minor comments:

1. Revise the typo “fist” on Page 5.

We have modified this accordingly.

2. The scale bars are randomly present, so please make them consistent for the figures.

We have modified them accordingly.

3. Figure 4C is confusing at first look. Colors can be represented in a legend.

We have modified this accordingly.

Reviewer #2 (Remarks to the Author):

Reviewer Comments to Author

This paper conveys a fabrication method to guide the morphogenesis of cells in vitro by imposing geometrical cues inside hydrogel using photo-sensitive polymers and two-photon mediated printing. This research is distinguished from the current printing methods where fabricated structures are confined to initial designs. Furthermore, this paper presents the versatility of the developed live printing technique, especially applications in guiding the branch-like structure of the intestine and respiratory tree. This paper will give a lot of inspiration for future researchers in this field, therefore can be recommended for publication with some major comments for authors to carefully consider.

Major comments

1. For Figure 2c-f, the reviewer is wondering what kind of dynamic hurdles the authors especially intended to mimic, and what kind of adaptation the cancer cells showed exactly when encountering the bioprinted environment. Please add some clear explanations and references if necessary.

The experimental setting reproduced two different kinds of constrains that caused the cancer organoids to grow through 1) multiple cell migration (pillars located at higher distance from each other) and 2) single cell migration (pillars locate in close proximity to each other). This opens the possibility to test how cancer cell can migrate from organoids and in 3D, mimicking different possible migration mechanisms observed already in vivo and in vitro. We have better clarified this point and included new references in the revised manuscript.

2. In this manuscript, the most extended observation timepoint after bioprinting is day 14. However, morphogenesis processes in vivo generally take a longer time than that, so the reviewer recommends the authors provide long-term (i.e., 4 weeks) results of the culture.

We understand the point raised by the reviewer. Regarding the organotypic culture of mouse fetal lung epithelium, we would like to highlight that our study is focused on the pseudoglandular stage of lung development (E9.5–E15.5), i.e. the stage when the airway tree is shaped through branching morphogenesis. We used E12.5 mouse embryos to isolate mesenchyme-free epithelium rudiments and cultured them up to 3-4 days, maintaining

branching morphogenesis. This period of culture covers the remaining period of pseudoglandular lung development in vivo. It is, therefore, not relevant to extend our culture beyond a time interval of 4 days.

3. Basically, it is more logical to give the readers a quantified accordance rate between the intended design and the bioprinted structure to demonstrate that the 3D structures were faithfully reproduced as designed, for example, in Figure 3a.

Figure 1, Figure 4, Suppl. Fig. S3 and Suppl. Fig. S5 present detailed reproduction of the desired design. Moreover, to guide the reader we have designed new cartoons to better explain the proposed experimental design, such as in **New Fig. 1a** and **New Fig. 5c**

New Fig. 1a

New Fig. 5c

4. In Figure 3d and Supplementary S5d, the authors presented the quantification data of the central and branched area covered by mSIOs. Please define what SIO1 to SIO6 means in the main manuscript.

We have now included this information in the main text of the new version of the manuscript.

5. The reviewer thinks the readers would expect to see how the mSIOs adapted to both circular central parts and the branched sides. Please consider presenting Supplementary Figure S5b and S5d in the main figure, for example, following Figure 3c-d.

We have modified the **new Fig. 4** (below), including the data presented in **new Suppl. Fig. S5**.

New Fig. 4

New Suppl. Fig. S5

6. Continuing on comment 5, please give some explanation about why the area occupied by the mSIOs are largely different from SIO1 to SIO6 (especially in the branch) and why the plots are not continuous through day 0 to 9 in Figure S5d. It would be great if the authors can present full images of Figure 3b or Figure S5b to show the readers that the adaptation of mSIOs occurred in every branch.

We would like to thank the reviewer for this comment. Regarding the area occupied by the mSIOs, it is strictly correlated to the starting size of the organoids. The split is done by manual disaggregation by shear stress, therefore at day 1 post seeding we have different sizes in culture. Therefore, at different time points some organoids need to grow longer to occupy the available volume defined by the constrain.

We have also performed new experiments to acquire full-size images showing the invasion of multiple crypts at different levels (Z-planes). **New Fig. 4c**, and **new Suppl. Fig. S5d** show the mSIOs adapting in 3 different branches of our design over a period of 7 days post-printing, as reposted below for the reviewer's convenience.

New Fig. 4c

New Suppl. Fig. S5d

7. The reviewer would like to kindly ask the authors to state how long does it take to print the 3D structures in the manuscript.

The printing speed depends on the microscope system being used and we can produce a 1 mm³ volume of cross-linked hydrogel in 30 minutes. The overall printing time depends on the printed structure (multiple ROIs can be printed simultaneously). We have now included this information in each section.

Minor comments

1. Please carefully check if there is Figure 1i (page 4, line 18).

New Figures were added and/or changed, and manuscript was modified accordingly.

2. Figure 1 legend: i, ii, iii are not labeled in the figure, and please check if the explanation of legend e) is appropriate. There is no scale bar in f.

Legend was properly labelled and scale bar in F was added.

3. Suppl. Fig. S1a: The blue and green letters are difficult to read.

A dark fill was added to the text.

4. Please correct the typo (stare→started) on page 5, line 24.

Typo was amended.

5. Figure 2 legend: There is an unnecessary bracket in the legend of d.

Typo was amended.

6. Suppl. Fig. S4a: Please add arrows to indicate the pillars.

Arrows were added to indicate the 4 interacting pillars.

7. Figure 3 legend: Please carefully check if the Suppl. Table 1 and 2 are correctly cited. The reviewer would recommend adding an appropriate, detailed statistical analysis part in the supplementary document if needed. It seems the legend d is not a complete sentence. Please carefully revise it.

We correctly cited Suppl. Tables S1 and S2, updated the following Tables and attached into the Supplementary Material.

8. Suppl. Fig. S5e: It is difficult to read the blue letters.

A dark fill was added to the text.

9. F-actin (ACT) is not used in Fig. 4b. Please carefully correct the manuscript (page 7, line 24).

We thank the reviewer for pointing out this error, which we amended accordingly in the text.

10. Please clearly explain what kind of lung epithelial cell was used in Figure 4d-f in the main manuscript.

The lung epithelial tissue used in Figure 4d-f was derived from pseudoglandular stage lungs of E12.5 mouse embryos (obtained through time-mated pregnancies). The lungs were isolated from the embryos, washed with PBS and incubated with 8U/mL Dispase for 2 min at room temperature. Then, mesenchyme was mechanically removed with tungsten needles and distal tissue including at least one bud tip was isolated and embedded in Matrigel drop to allow for 3D culture. We have highlighted this in the main manuscript and in the material and method section.

11. The reviewer thinks there's a redundancy in the paragraph starting with 'Then, we investigated the possible biological effect of live bioprinting' on page 8. Please concise this part or rephrase.

We rephrased this sentence.

Reviewer #3 (Remarks to the Author):

The authors present a novel and interesting way to use 2P crosslinking to allow for the printing of 3D structures within pre-existing hydrogel-based matrices (i.e. Matrigel) through the diffusion of the photo-sensitive polymer in the pre-existing hydrogel structure. This allows for alterations over time within the 3D culture environment to steer and/or influence the development of organoid cultures. The methodology appears to be sound and the use of a range of different cell types/ organoids is a strong aspect of the manuscript.

The presented concept is derived from the previous work of the authors (reference 9 in the manuscript). In my view, the main novelty of the present work is that the photo sensitive polymer is not mixed with the Matrigel, but, after seeding of the cells, allowed to diffuse in the cell-laden structures. This could then allow modifications in, e.g. mechanical, properties also later to be added to the hydrogel structures. This novelty is well visualized in Figure 1A. Subsequently, number of potential applications of the technology are explored using different “organ-like” and organoid structures.

While the concept is very interesting and will provide an additional tool in the field of bioprinting, the technological step is somewhat incremental from the authors previous work (especially regarding reference 9). The provided examples are “very nice”, but not “overwhelming”. In view of this, I do feel the work should be published, but probably in a more specialized journal

We would like to thank the reviewer for emphasizing the major technological advance which made the experiments presented in this manuscript possible, and for agreeing that the example applications provided are very nice. These examples will be of interest to scientists working on neurogenesis, lung development, liver regeneration, intestinal patterning, cancer metastasis, chemical engineering, stem cells and mechanobiology. As such, this is truly multi-disciplinary work which we anticipate will be highly cited by authors in multiple fields and the ease of use of the technology we describe will enable it to be adopted by many groups quickly.

We must also emphasize additional points of novelty and experimental hurdles we overcame in this manuscript:

- The ability to change geometric constraints allows analysis of different cellular mechanisms. For example, the cancer organoids we study first migrate as aggregates through wide pillars but can then switch to migrating through thin slits which only allow individual cell migration.
- Spatial and temporal control over imposed geometries means specialized cell types can be allowed to emerge in established culture conditions before imposing constraints. We demonstrate this with the outgrowth of neurites.
- The ability to induce patterning events with physical structures increases the predictability of otherwise stochastic culture systems. Residual variability, as we observe in the induction of lung branching morphogenesis, provides insights into the requirements for predictable morphogenesis in vivo.

Some more minor comments and suggestions

1. In the Abstract “2P” is not explained

We have modified this accordingly in the abstract.

2. Use of “4D” in the field is not very defined. The use of “live 4D” does also not really add. I would suggest to use the term “spatio-temporal” as this is more descriptive.

We have modified this accordingly and also based on the suggestions of Reviewer #1.

3. What do the authors mean when using “organ-like”?

It was a general term to include both organotypic and organoid cultures as reported by others. We have stated this clearly into the new version of the manuscript.

4. The meaning of the following sentence is unclear (consider rephrasing): “Newly fabricated hydrogels can be tailored to the specific culture system requests and at the desired culture time, matching the spatiotemporal requirement to control 3D organ-like cultures.”

We have modified this to make the sentence clearer.

5. How does the presence of the organoids/cells influence the resolution?

In our experiments we do not find any alteration in the printing resolution when the organoids/cells are present into the solid hydrogel. We have now stated this in the manuscript. Also, based on the comments raised from the Reviewer #2, we have now included in each figure the original design and the final achieved design of the printed structures to clarify the ability to generate structures with the desired shape.

6. Did the authors also evaluate the mechanical characteristics of the newly formed structures (beyond Fig 1h)? This could really provide this insights on how this technology could mimic the mechanically changing micro environment.

Since we show that the hydrogel stiffness is not really affected by the presence of the solid hydrogel in which they are printed, we know that hydrogels can be fabricated at the desired stiffness by modifying the laser power and/or the concentration of the polymers (as fully characterized in the previous *Nat Biom Eng* paper). The Reviewer's suggestion is really interesting, and it is something on which we are currently working in an ongoing new project.

7. The intestinal organoids were only followed for a short period of time. Probably the lumen would quickly be overgrown due to the lack of a mechanism to remove the dislodged cells (see also work of Lutolf Lab). Have the authors considered to implement this into a (micro) perfusable system?

The reviewer highlights a good point, as in the current system we cannot "wash" away from the Matrigel/Gel the differentiated villi cells of the mSIOs. For this reason, these experiments were performed for a reduced period of time of up to 10/14 days post-split to prevent the organoids from "suffocating" from discarded cells. The reviewer is right and a microfluidic system is something we are implementing in a follow-up new work.

8. Regarding the budding of the lung organoids, while budding it self happens spontaneously (See also fig 4d); what is the main advantage of being able to control this? Do the buds at the later stages of culture fuse again (when they have grown around the pillars)?

We agree with the reviewer on this point, as the budding itself happens spontaneously in vivo as well as in our ex vivo settings. From a technological point of view, we propose a

system to achieve a model of stereotyped branching that would provide several benefits for developmental biologists studying branching morphogenesis, including:

- the possibility to standardize the ex vivo culture of branching tips;
- the possibility of investigating the interaction with extrinsic cues such as the stiffness or the chemical composition, for example, to mimic a specific ECM component. This is particularly interesting as the extracellular environment is increasingly recognized as key in regulating developmental processes of branching morphogenesis¹;
- the possibility to accurately investigate the temporal dynamics of branching, for example by analyzing the time of cleft depth formation around pillars of precise dimensions.

We are also confident this model fairly recapitulates the branching and the established culture conditions do not allow the fusion of tips while in culture. We have included a **new Suppl. Video 5** that shows a 160 μm Z-stack of an immunofluorescence staining of a representative lung epithelium sample that underwent branching in the presence of two pillars and cultured for additional 3 days. In the video, the tip and stalk regions can be appreciated by Sox9 (Cyan) and Sox2 (Red) immunostaining and no fusion of different tips have been observed, even after a prolonged period of culture, i.e. 3 days.

Bibliography

1. Uçar, M. C. *et al.* Theory of branching morphogenesis by local interactions and global guidance. *Nat. Commun.* **12**, (2021).

REVIEWER COMMENTS

Reviewer #4 (Remarks to the Author):

In their manuscript Urciuolo and colleagues introduce a hydrogel-in-hydrogel approach. Using two photon photopolymerization, the authors demonstrate patterns of adhesive or non-degradable PEG-based and gelatin-based hydrogels inside of a droplet of Matrigel. The process works via monomer diffusion through the Matrigel droplet followed by raster scanning to polymerize 3D patterns. The process is applied across a broad range of tissue, demonstrating 3D axon guidance in spinal cord explants, and geometric constraints in cancer spheroids, intestinal organoids, and human fetal hepatocyte organoids.

The following comments relate to the original comments from Reviewer #1, and whether the author's response adequately addresses the reviewer's concerns.

Overall, it is my opinion that the hydrogel-in-hydrogel approach described here represents a significant advance, enabling the writing of non-permissive elements into typically highly permissive Matrigel and the assays performed are excellent demonstrators of the different applications of this approach. I believe that some additional analysis is required to provide quantifiable data on the reproducibility of directing the budding of lung organoids around pillars (see point 5). Pending this analysis, I recommend acceptance of the manuscript. Specific responses to major comments are provided below:

1. The authors have removed the term 4D bioprinting, which I agree is a somewhat vague term, and are now using 'hydrogel-in-hydrogel live bioprinting'. I think this is an excellent and descriptive term for this process.
2. While on the surface, Fig 5 in the authors' prior Nat Biomed. Eng., 2020 paper bares similarity to Figs. 4 and 5 in this manuscript, the similarity is superficial. I agree with the authors that addition of monomers to a pre-gelled Matrigel uniquely enables polymerization at later timepoints. This is particularly well demonstrated in the new data in Figure 3 wherein a second wall of pillars are printed around a growing cancer organoid to pin its growth.
3. Reviewer #1 flagged potential toxicity of monomers and photoinitiators that are introduced at a later stage. As the authors describe, the short (~15 minute) incubation of monomers and rinsing post-printing limits exposure. While viability data in Fig 2b-d would be enhanced with an ethidium homodimer 'dead' stain, the data in Fig S3e and Fig S5b is highly convincing that cell death is limited, if at all, present due to printing.
4. I believe the additional information provided by authors more accurately describes the method used to identify when and how to position the pillars.
5. Reviewer #1's concern over reproducibility is warranted due to high variability seen in quantified morphological data in Fig 4d-g and a lack of quantification in Fig. 5. In general, I do not see some variability as an issue as this as variation is inevitable in many organoid experiments, and the manuscript does not specifically claim to enhance reproducibility. However, I do not think that the response 'reproducibility is about 50%' is sufficiently rigorous; only a few lung organoids are shown in the main text at different magnification with no quantitative discussion on how reproducible this result is. I suggest a more quantitative analysis of lung branching and providing numbers to back up the about 50% success rate.
6. The authors' response to the question of print shapes is thorough.

7. To demonstrate printing at time $t > 0$, The authors now print an additional wall of pillars to further constrain the growth of a cancer organoid after it has grown beyond the first wall of pillars.

8. I consider the authors' response to the question of the benefits of printing at $t > 0$ is acceptable, although demonstrating this for the branching lung organoids would be particularly compelling.

9. The new supplementary data clearly demonstrates hollow structures being formed.

Minor comments:

Line 545 - The manuscript here still refers to '4D printing' - should use term 'live bioprinting' for consistency.

Line 752 - typo - should read "Organoid"

Line 755 - typo - should read 'scale bar'.

The following is an assessment of the comments provided by Reviewer #3 and the associated responses provided by the authors.

As indicated by Reviewer #3, the work is indeed novel and interesting, and I would assess the impact and quality of the work as 'very good' - or excellent. There are no other methods that polymerize a non-permissive hydrogel inside of a permissive gel like Matrigel, and this offers a fundamental new ability to dynamically alter the growth landscape of growing cells, organoids, and tissues. I do agree with the reviewer that the presented results are not 'overwhelming', but it is my belief that they will be broadly appreciated by the tissue engineering, biofabrication, and organoid development fields alike.

With regards to the specific minor concerns provided by Reviewer 3, I believe that the authors have acceptably responded to the minor concerns with minor textual edits. Of note, I agree with the authors that a more detailed analysis of stiffness in their IPNs or longer term studies that would require a microfluidic chip are out of scope of the current project.

Reviewer #5 (Remarks to the Author):

This study utilizes two-photon crosslinking principles to print hydrogels with guiding cues inside hydrogels that support organoid growth to guide cell and organoid growth in 3D. The authors have demonstrated the potential of this strategy to control axon unidirectional growth and cancer cell migration. Although this study is an incremental step of their previous work as already stated by other reviewers, it deserves to be published given the new shown examples and applications.

Before publication, the authors should clarify and revise some definitions. First, the term "bioprinting" refers to printing of living cells and not to the printing of 3D structures around or surrounding cells. Thus, I suggest changing the term "bioprinting" to "hydrogel-in-hydrogel printing" as already used by the authors several times throughout the manuscript. Next, the term "live" should be omitted given that all 3D printing procedures (so far) are performed in real time. The title of the article should be amended accordingly to "Hydrogel-in-hydrogel printing for structural cell guidance and dynamic control of organoid and organotypic cultures".

Dear Editor,

We have improved our manuscript with new experiments and new figures. Here we present a detailed point-by-point rebuttal to the final issues raised by the reviewers.

REVIEWER COMMENTS

Reviewer #3 (Remarks to the Author)

The following is an assessment of the comments provided by Reviewer #3 and the associated responses provided by the authors.

As indicated by Reviewer #3, the work is indeed novel and interesting, and I would assess the impact and quality of the work as 'very good' - or excellent. There are no other methods that polymerize a non-permissive hydrogel inside of a permissive gel like Matrigel, and this offers a fundamental new ability to dynamically alter the growth landscape of growing cells, organoids, and tissues. I do agree with the reviewer that the presented results are not 'overwhelming', but it is my belief that they will be broadly appreciated by the tissue engineering, biofabrication, and organoid development fields alike.

With regards to the specific minor concerns provided by Reviewer 3, I believe that the authors have acceptably responded to the minor concerns with minor textual edits. Of note, I agree with the authors that a more detailed analysis of stiffness in their IPNs or longer term studies that would require a microfluidic chip are out of scope of the current project.

We would like to thank the reviewer 3 for the approval of our revised manuscript.

Reviewer #4 (Remarks to the Author)

In their manuscript Urciolo and colleagues introduce a hydrogel-in-hydrogel approach. Using two photon photopolymerization, the authors demonstrate patterns of adhesive or non-degradable PEG-based and gelatin-based hydrogels inside of a droplet of Matrigel. The process works via monomer diffusion through the Matrigel droplet followed by raster scanning to polymerize 3D patterns. The process is applied across a broad range of tissue, demonstrating 3D axon guidance in spinal cord explants, and geometric constraints in cancer spheroids, intestinal organoids, and human fetal hepatocyte organoids.

The following comments relate to the original comments from Reviewer #1, and whether the author's response adequately addresses the reviewer's concerns.

Overall, it is my opinion that the hydrogel-in-hydrogel approach described here represents a significant advance, enabling the writing of non-permissive elements into typically highly permissive Matrigel and the assays performed are excellent demonstrators of the different applications of this approach. I believe that some additional analysis is required to provide quantifiable data on the reproducibility of directing the budding of lung organoids around pillars (see point 5). Pending this analysis, I recommend acceptance of the manuscript. Specific responses to major comments are provided below:

We would like to thank the reviewer 4 for his substantial approval of our revised manuscript. We agree with him that further work was required in regards of lung bifurcation data.

1. The authors have removed the term 4D bioprinting, which I agree is a somewhat vague term, and are now using 'hydrogel-in-hydrogel live bioprinting'. I think this is an excellent and descriptive term for this process.

Thank you.

2. While on the surface, Fig 5 in the authors' prior Nat Biomed. Eng., 2020 paper bares similarity to Figs. 4 and 5 in this manuscript, the similarity is superficial. I agree with the authors that addition of monomers to a pre-gelled Matrigel uniquely enables polymerization at later timepoints. This is particularly well demonstrated in the new data in Figure 3 wherein a second wall of pillars are printed around a growing cancer organoid to pin its growth.

Thank you.

3. Reviewer #1 flagged potential toxicity of monomers and photoinitiators that are introduced at a later stage. As the authors describe, the short (~15 minute) incubation of monomers and rinsing post-printing limits exposure. While viability data in Fig 2b-d would be enhanced with an ethidium homodimer 'dead' stain, the data in Fig S3e and Fig S5b is highly convincing that cell death is limited, if at all, present due to printing.

Thank you.

4. I believe the additional information provided by authors more accurately describes the method used to identify when and how to position the pillars.

Thank you.

5. Reviewer #1's concern over reproducibility is warranted due to high variability seen in quantified morphological data in Fig 4d-g and a lack of quantification in Fig. 5. In general, I do not see some variability as an issue as this as variation is inevitable in many organoid experiments, and the manuscript does not specifically claim to enhance reproducibility. However, I do not think that the response 'reproducibility is about 50%' is sufficiently rigorous; only a few lung organoids are shown in the main text at different magnification with no quantitative discussion on how reproducible this result is. I suggest a more quantitative analysis of lung branching and providing numbers to back up the about 50% success rate.

We thank the reviewer for his suggestion. To address this issue, we performed new experiments on different biological replicates (as per mouse fetal lung rudiments) to increase data numerosity and consistency. In these experiments, we printed pillars in front of non-bifurcated lung tips. Out of 13 analysed rudiments, 9 showed proper pillar-driven tip bifurcation. This is because in 9 samples the tip grew in the correct direction towards the printed pillar and the contact tip-pillar happened. In the remaining 4 samples, the tip either never grew, therefore never touched the pillar, or it grew vertically surpassing the height of the pillar, so no contact tip-pillar happened. In 48 hours post-printing, all 9 samples in which the tip contacted the pillar underwent pillar-driven bifurcation. We can therefore conclude that the efficiency of hydrogel-in-hydrogel bioprinting to control lung tip bifurcation is 70%, while the efficacy of pillar-driven bifurcation upon tip-pillar contact is 100% in our analysed samples.

Based on this new set of experiments, we could measure the dynamic of pillar-driven tip bifurcation during rudiments culture, by measuring the angle between the pillar and the growing tip upon pillar touch. More in detail, once the growing tips reached the printed pillar (t0) we calculated the mean of right and left angles of tip bifurcation of 8 independent samples on every hour for 24 hours. The angles were measured between each tangent to the pillar and the parallel direction of bifurcating tip.

We are now showing these data in the **New Fig.5e** (also presented below). Accordingly, in the revised manuscript we also modified the text on page 9 in the result section, and on page 17 in the methods section.

6. The authors' response to the question of print shapes is thorough.

Thank you.

7. To demonstrate printing at time $t > 0$, The authors now print an additional wall of pillars to further constrain the growth of a cancer organoid after it has grown beyond the first wall of pillars.

Thank you.

8. I consider the authors' response to the question of the benefits of printing at $t > 0$ is acceptable, although demonstrating this for the branching lung organoids would be particularly compelling.

We would like to clarify to the reviewer that in all lung bifurcation experiments (including new experiments presented here) we extracted the lung rudiments from mouse fetuses and plated them in Matrigel, culturing them for 24 hours before proceeding with 3D printing. After 24 hours we equilibrated the Matrigel with liquid PEG gel and proceeded with printing and analyses for a further 72 hours post printing. We can therefore consider $t > 0$ as all printing happened after 24 h of culture. We clarified this in the main text, results section, page 8.

9. The new supplementary data clearly demonstrates hollow structures being formed.

Thank you.

Minor comments:

Line 545 - The manuscript here still refers to '4D printing' - should use term 'live bioprinting' for consistency.

We changed this to live bioprinting, as suggested.

Line 752 - typo - should read "Organoid"

We amended the typo, as suggested.

Line 755 – typo – should read 'scale bar'.

We amended the typo, as suggested.

Reviewer #5 (Remarks to the Author)

This study utilizes two-photon crosslinking principles to print hydrogels with guiding cues inside hydrogels that support organoid growth to guide cell and organoid growth in 3D. The authors have demonstrated the potential of this strategy to control axon unidirectional growth and cancer cell migration. Although this study is an incremental step of their previous work as already stated by other reviewers, it deserves to be published given the new shown examples and applications.

Before publication, the authors should clarify and revise some definitions. First, the term "bioprinting" refers to printing of living cells and not to the printing of 3D structures around or surrounding cells. Thus, I suggest changing the term "bioprinting" to "hydrogel-in-hydrogel printing" as already used by the authors several times throughout the manuscript. Next, the term "live" should be omitted given that all 3D printing procedures (so far) are performed in real time. The title of the article should be amended accordingly to "Hydrogel-in-hydrogel printing for structural cell guidance and dynamic control of organoid and organotypic cultures".

We would like to thank the reviewer 5 for the approval of our revised manuscript. We also understand the concern regarding the title of the manuscript and the definition of the technology. However, we decided to use the term "bioprinting" since hydrogels are printed in presence of cells in the system. This is related to print in proximity of organoids/organotypic cultures, or within cytostructures of organotypic cultures (please refer to Fig.2 for the latter where cells were embedded within the printed structures).

Regarding the term "live", this means that bioprinting is associated to live imaging. Indeed, each single hydrogel structure was printed upon and during live imaging observation with the 2P-microscope.

For this reason, we still think that the title of the manuscript and the definition of the technology is appropriate and not misleading, as it is thoroughly described and explained through the manuscript.

REVIEWERS' COMMENTS

Reviewer #4 (Remarks to the Author):

The authors have included new data demonstrating the reproducibility of directed tip bifurcation of fetal lung organoids. In doing so, they have addressed all of my remaining concerns, and I recommend the publication of this manuscript.

Dear Editor,

Here we present a detailed point-by-point rebuttal to the final Reviewer's comments.

REVIEWER COMMENT

Reviewer #4 (Remarks to the Author)

The authors have included new data demonstrating the reproducibility of directed tip bifurcation of fetal lung organoids. In doing so, they have addressed all of my remaining concerns, and I recommend the publication of this manuscript.

We would like to thank the Reviewer 4 for the approval of our revised manuscript.